# The vaccinia chondroitin sulfate binding protein drives host membrane curvature to facilitate fusion

Laura Pokorny [ID][1,2], Jemima J Burden [ID][1], David Albrecht [ID][2], Rebecca Bamford[1,2], Kendra E Leigh [ID][3,4], Pooja Sridhar [ID][5], Timothy J Knowles [ID][5], Yorgo Modis [ID][3,4] & Jason Mercer [ID][1,2✉]

## Abstract

**Cellular attachment of viruses determines their cell tropism and species specificity. For entry, vaccinia, the prototypic poxvirus, relies on four binding proteins and an eleven-protein entry fusion complex. The contribution of the individual virus binding proteins to virion binding orientation and membrane fusion is unclear. Here, we show that virus binding proteins guide side-on virion binding and promote curvature of the host membrane towards the virus fusion machinery to facilitate fusion. Using a membrane-bleb model system together with super-resolution and electron microscopy we find that side-bound vaccinia virions induce membrane invagination in the presence of low pH. Repression or deletion of individual binding proteins reveals that three of four contribute to binding orientation, amongst which the chondroitin sulfate binding protein, D8, is required for host membrane bending. Consistent with low-pH dependent macropinocytic entry of vaccinia, loss of D8 prevents virion-associated macropinosome membrane bending, disrupts fusion pore formation and infection. Our results show that viral binding proteins are active participants in successful virus membrane fusion and illustrate the importance of virus protein architecture for successful infection.**

**Key words** Poxvirus; Membrane Bending; Glycosaminoglycans; Virus Entry; Membrane Fusion
**Subject Categories** Membranes & Trafficking; Microbiology, Virology & Host Pathogen Interaction; Structural Biology

## Introduction

Infectious mature virions (MVs) of all poxviruses, including variola (the causative agent of smallpox), monkeypox and the smallpox vaccine, vaccinia virus (VACV), encode 4 distinct binding proteins and an 11-protein entry fusion complex (EFC) (Moss, 2007). Each of the 11 EFC proteins is needed for membrane fusion competence (Laliberte et al, 2011; Moss, 2012), while the 4 reported binding proteins A26, A27, D8 and H3 differentially contribute to MV attachment (Hsiao et al, 1998, 1999; Chung et al, 1998; Lin et al, 2000; Chiu et al, 2007). The A26 protein has been shown to mediate virus binding to extracellular laminin and to serve as a fusion repressor (Chiu et al, 2007; Chang et al, 2019, 2012). A27, H3 and D8 are glycosaminoglycan (GAG) binding proteins reported to interact with heparin sulfate (A27, H3) or chondroitin sulfate (D8), respectively (Hsiao et al, 1999; Chung et al, 1998; Lin et al, 2000). While soluble A27, H3 and D8 each interfere with virion binding, only deletion of D8 was shown to significantly impact GAG-mediated cell surface attachment of MVs (Chung et al, 1998; Lin et al, 2000; Hsiao et al, 1999). Collectively, various reports indicate that VACV GAG usage varies with cell type, virus strain and experimental condition (Bengali et al, 2010, 2013; Carter et al, 2005; Whitbeck et al, 2009). Due to this complexity, the intricacies and individual contribution of these proteins to VACV MV cell surface binding remain poorly understood.

We recently showed that the VACV membrane is organized into distinct functional domains; with EFCs polarized to the tips of virions and binding proteins A27 and D8 relegated to the sides of virions (Gray et al, 2019). A27-dependent EFC polarization was found to be critical for tip-oriented fusion, and localization of D8 to correlate with a side-on virion binding bias (Ewers et al, 2010; Gray et al, 2019). This prompted the questions: Which of the four viral binding proteins contribute to VACV binding orientation and how does this correlate or contribute to tip-oriented fusion?

## Results

### A cell-derived membrane bleb model system enables the study of VACV binding and fusion

Traditional membrane model systems such as liposomes and giant unilamellar vesicles (GUVs) have been invaluable for investigating binding, uptake and fusion of several viruses (Ewers et al, 2010; Nikolaus et al, 2010; Rydell et al, 2013; Ho et al, 2016). However, as VACV binding requires multiple cell surface proteoglycans, investigation in these lipidic systems is inefficient and of

[1]Institute of Microbiology and Infection, School of Biosciences, University of Birmingham, Birmingham B15 2TT, UK. [2]MRC-LMCB, University College London, London WC1E 6BT, UK. [3]Molecular Immunity Unit, Department of Medicine, University of Cambridge, MRC Laboratory of Molecular Biology, Francis Crick Avenue, Cambridge Biomedical Campus, Cambridge CB2 0QH, UK. [4]Cambridge Institute of Therapeutic Immunology & Infectious Disease (CITIID), University of Cambridge School of Clinical Medicine, Cambridge CB2 0AW, UK. [5]School of Biosciences, University of Birmingham, Birmingham B15 2TT, UK. ✉E-mail: j.p.mercer@bham.ac.uk

questionable biologically relevance (Schmidt et al, 2013). To this end we established a minimal model system based on cell-derived membrane blebs (Biro et al, 2013) to facilitate the quantification of large numbers of binding events in both fluorescence- and electron microscopy (EM)-based imaging experiments (Figure EV1). Using a fluorescent-recombinant VACV (EGFP-A4 (Mercer and Helenius, 2008)) we found that blebs could be used for analysis of virus binding (Fig. 1A), and when coupled with our established octadecylrhodamine (R18)-dequenching assay (Gray et al, 2019; Schmidt et al, 2013), pH-dependent hemi-fusion (Fig. 1B). Use of the fusion neutralizing anti-L1 antibody (7D11) confirmed that the observed low pH-mediated dequenching was due to virus-bleb fusion and not non-fusogenic dye transfer (Schmidt et al, 2013; Doms et al, 1990; Fig. 1B).

Having shown that our bleb model system supports virus binding and fusion, we turned our attention to accurately identifying and quantifying VACV binding orientation during fusion. Using scanning- and transmission- EM (TEM) respectively we recently reported that VACV virions preferentially bind on their sides and undergo low-pH dependent fusion at their tips (Gray et al, 2019). This suggested to us that membrane-bound virus may undergo reorientation in response to low pH bringing the virion tips into contact with the cell membrane. To test this, recombinant virions harboring a mCherry-tagged core protein and an EGFP-tagged lateral body protein (mCherry-A4 F17-EGFP) (Schmidt et al, 2012; Gray et al, 2016) were loaded with R18 dye, bound to blebs and subjected to neutral (7.4) or low (5.0) pH treatment. Blebs bound with a single virion were visualized by Structured Illumination Microscopy (SIM) (Fig. 1C). Using core elongation and lateral body separation as parameters, virion binding orientation was plotted as side/tip ratio (Fig. 1C, D). The pH 5.0 samples were also scored for virion hemi-fusion as evidenced by R18-dequeching into the bleb membrane (Fig. 1C, D). In all cases virions showed preferential side binding with no significant changes in binding orientation observed upon low pH treatment or subsequent hemi-fusion (Fig. 1D). Intriguingly, these results suggested that despite the localization of the EFC to virion tips, VACV does not need to be bound on its end to undergo fusion with host membranes.

## VACV binding induces membrane invagination under low pH conditions

To reconcile this, we used TEM to further investigate the binding interaction between virions and blebs at pH 7.4 and pH 5.0. As expected, under both conditions, virions bound preferentially on their sides. While we observed no discernible changes in the underlying bleb membrane at pH 7.4 (Fig. 1E; top row), at pH 5.0 invagination of the bleb membrane occurred specifically under bound virions (Fig. 1E, bottom row). Quantification indicated that at pH 7.4, only 7.1% of virions were found in invaginations between 11 and 72.2 nm deep (average depth = 36 nm), while at pH 5.0, 46% of virions resided in invaginations ranging from 13 to 144 nm deep (average depth = 60 nm) (Fig. 1F). When the pH 5.0 samples were scored by virion binding orientation, 48% of side bound virions were found in invaginations as opposed to 15% of tip bound virions (Fig. 1G). Collectively these results confirm that VACV binds to host membranes in a side-on orientation, indicate that binding orientation is not altered by low pH or the induction of hemi-

fusion, and suggest that when bound on their side VACV virions manipulate the curvature of cellular membranes in response to low pH.

Since the VACV membrane is organized into functional binding and fusion domains (Gray et al, 2019) we reasoned that VACV binding protein(s) may be responsible for the induction of host membrane curvature. Having shown that VACV binding proteins A27 and D8 reside at the sides of virions (Gray et al, 2019), we first investigated the localization of the two remaining binding proteins, A26 and H3 (Chiu et al, 2007; Lin et al, 2000). For this, we generated recombinant EGFP core (EGFP-A4) viruses that express HA-tagged versions of A26 or H3. Using SIM and the single-particle averaging software VirusMapper (Gray et al, 2016), we mapped the localization of A26 and H3 on VACV virions. Virions immuno-stained for A27 and D8 were included for comparison. VirusMapper models and determination of the polarity factor indicated that like A27 and D8, A26 and H3 are largely relegated to the sides of virions (Figs. 2A and EV2A). We confirmed these results using Stochastic Optical Reconstruction Microscopy (STORM), a higher resolution technique that also suggested that A26 and H3 reside in the VACV membrane as clusters akin to A27 and D8 (Figure EV2B; (Gray et al, 2019)).

## VACV A26, D8 and H3 surface proteins contribute to virus binding affinity and orientation

To gain a comprehensive assessment of the relative contribution of these four proteins to VACV binding affinity and orientation we generated EGFP-core versions of inducible A27 and H3 VACV recombinants, as well as VACV strains in which A26 or D8 were deleted. We compared the binding capacity of WT virions to those lacking A26 (ΔA26), A27 (A27-), D8 (ΔD8) or H3 (H3-). For this, equal numbers of virions were adsorbed to cells and binding quantified by flow cytometry for EGFP signal. Loss of A27 showed no impact on VACV binding, while loss of A26, D8 or H3 reduced VACV binding by 66, 47, and 24%, respectively (Figure EV2C). Results from our bleb binding orientation assay showed similar results: A27- virions preferentially bind side-on like WT virions, while a significant proportion (2.2 fold) of ΔA26, ΔD8 and H3-virions adapt a tip-on binding orientation (Fig. 2B). These results suggest that despite its ability to bind heparin sulfate (Chung et al, 1998; Hsiao et al, 1998), A27 plays no apparent role in VACV binding. They also demonstrate that A26, D8 and H3 each contribute to virion side-on binding orientation, and that A26 contributes most to cell surface attachment, followed by D8 and H3 (Figure EV2C).

## VACV D8 is required for low-pH dependent membrane invagination activity

Given these results, it reasons that A26, D8, H3, or a combination thereof contribute to the observed low pH-dependent membrane invagination. Using TEM we compared the membrane invagination capacity of WT, ΔA26, ΔD8 and H3- virions on blebs at pH 5.0. While we could readily observe WT, ΔA26 and H3- virions residing in invaginations, that vast majority of ΔD8 virions were associated with unaltered bleb membranes (Fig. 2C) resembling WT virion binding at pH 7.4 (Fig. 1E; top row). Quantification showed that the percentage of WT, ΔA26 and H3- virions found within

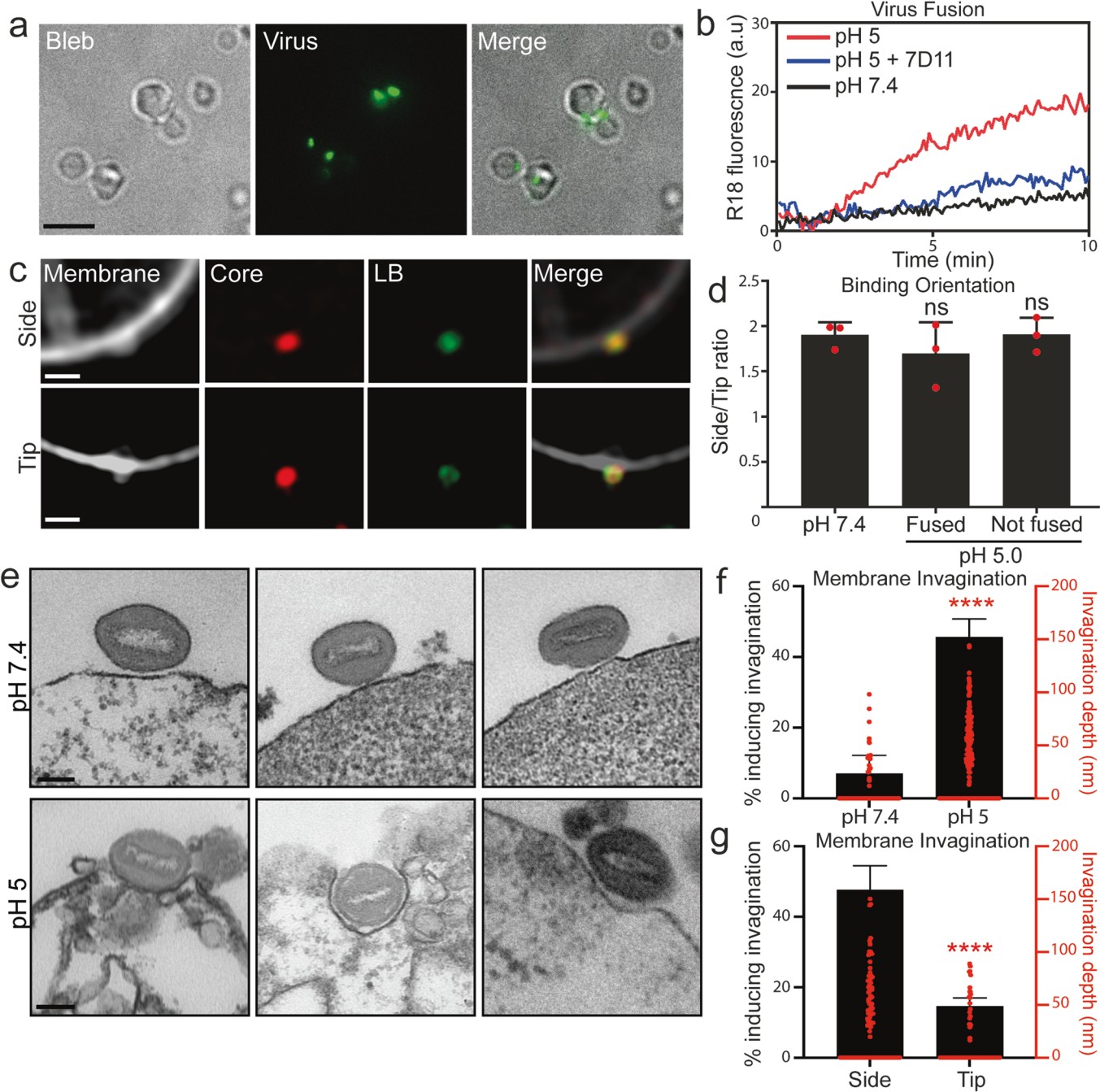

**Figure 1. VACV binds to, fuses with and mediates curvature of membrane blebs.**

(A) Blebs incubated with A4-EGFP virus at 4 °C for 1 h, washed and imaged. Scale bar, 5 µm. (B) VACV hemifusion rates (R18 dequenching assay) with blebs at pH 7.4 or pH 5.0. 7D11 fusion neutralizing antibody was used as a positive control. Fluorescence was normalized to the initial value and fully de-quenched value upon TX-100 addition. (C) Examples SIM images of mCherry-A4 EGFP-F17 VACV bound to blebs in a side-on (upper row) and tip-on (lower row) orientation. Scale bar = 1 µm. (D) Quantification of side/tip binding ratio at different pH's and fusion states ($n = 50$ virions/replicate). (E) TEM images of VACV bound to HeLa cell derived blebs at pH 7.4 (top row) or pH 5.0 (bottom row). Scale bar = 100 nm. (F) Quantification of membrane invagination depth under virions at pH 7.4 and pH 5.0 ($n > 50$ virions/replicate). (G) Quantification of side-on vs. tip-on bound virion invagination ($n > 50$ virions/replicate). Data information: In (D, F, G), data are mean ± standard deviation (SD) of biological triplicates. Statistical analysis was performed using unpaired two-tailed $t$-tests (****$P < 0.0001$; ns, not significant ($P > 0.05$)). Source data are available online for this figure.

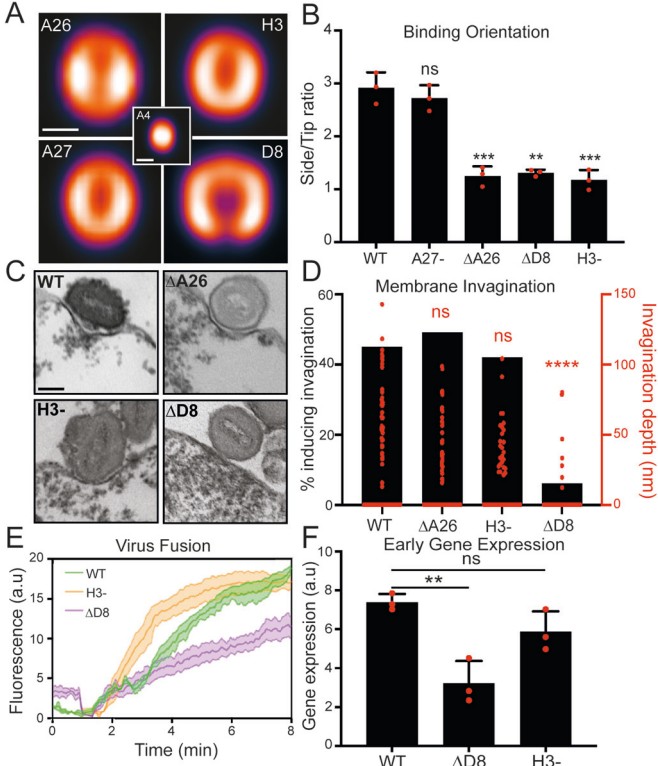

**Figure 2. VACV binding protein D8 is required for host membrane invagination and fusion.**

(A) VirusMapper localization models of VACV binding proteins. EGFP-A4 core (center) was used to correlate virion orientation. Models are representative of $n > 180$ virions. Scale bar, 200 nm. (B) Analysis of side/tip binding ratio of WT and binding protein mutants on blebs ($n = 50$ virions/replicate). (C) TEM images of WT and binding protein mutants bound to HeLa blebs at pH 5.0. Scale bar, 100 nm. (D). Quantification of percent invagination and invagination depth of binding protein mutants at pH 5.0 ($n > 50$ virions/mutant). (E) Hemifusion rates of WT, ΔD8 and H3- virions on HeLa cells using the R18 dequenching assay. (F) Comparison of WT, ΔD8 and H3- virus early gene expression at 2 hpi by RT-qPCR of early gene C11R. Data information: For (B, E), data are means ± SEM of biological triplicates and for (F) data are means ± SD. Statistical analysis was performed using unpaired two-tailed $t$ tests (****$P < 0.0001$; ***$P < 0.001$; **$P < 0.01$; ns, not significant ($P > 0.05$)). Source data are available online for this figure.

invaginations was similar, ranging between 42–49%, as opposed to only 6% of ΔD8 virions (Fig. 2D; black bars). This was consistent with a significant decrease in the average invagination depth of ΔD8 virions relative to WT, ΔA26 and H3- virions (Fig. 2D; red data points). Taken together, these results show that A26, H3 and D8 each contribute to VACV cell surface binding differentially. While all three proteins contribute to binding orientation to a similar degree, A26 appears to be most important for attachment, and D8 for a newly uncovered role in low-pH dependent membrane invagination.

## D8-mediated membrane invagination increases VACV fusion efficiency from macropinosomes

We next sought to determine the relevance of low pH dependent, D8-mediated membrane invagination. Given the distinct spatial

distribution of the viral binding proteins to the sides and fusion proteins to the tips of VACV virons (Gray et al, 2019), we hypothesized that this membrane "curving" activity of D8 could serve to bring the host membrane into contact with the virus fusion machinery under low pH conditions, such as those found in late macropinosomes from which VACV fuses (Townsley and Moss, 2007; Rizopoulos et al, 2015; Schmidt et al, 2013). If correct, we would expect VACV fusion activity to be impacted by the loss of D8. To investigate this, we compared WT, ΔD8 and H3- hemifusion rates on HeLa cells using our R18 dequenching assay. ΔD8 hemifusion was found to be reduced by 50% relative to WT and H3- viruses (Fig. 2E). For ΔD8, this result correlated with a 2.3-fold reduction in early gene expression—the earliest read-out of core entry into the cytoplasm—compared to WT and H3—as determined by real-time quantitative PCR (RT-qPCR) for the canonical early gene, C11R (Fig. 2F). These results demonstrate that loss of D8, and its low pH-dependent membrane invagination activity, impacts VACV fusion efficiency and the kinetics of early gene expression.

VACV can enter cells by direct fusion at the plasma membrane (Carter et al, 2005; Doms et al, 1990) or via low pH-dependent micropinocytosis (Mercer and Helenius, 2008; Schmidt et al, 2011; Huang et al, 2008). Therefore, we wanted to determine if D8-mediated invagination occurs at the limiting membrane of macropinosomes during VACV entry. As VACV resident time in macropinosomes is relatively short (Rizopoulos et al, 2015) making it difficult to capture in significant numbers, we took advantage of an A28- EFC mutant VACV which can undergo hemifusion, but not full fusion (Senkevich et al, 2004; Laliberte et al, 2011). Importantly, using EM we first confirmed that A28- virions induce pH-dependent invagination of blebs akin to WT virions (WT:40% vs A28-:42%) (Fig. 3A,B).

We then generated an A28- ΔD8 virus, confirmed that it did not express D8 (Figure EV2D) and used this in combination with a TEM horseradish peroxidase (HRP) entry assay to investigate the association of WT A28- ΔD8 virions with the macropinosome membrane. The HRP was used to increase the electron density within endosomes, making them easily visible by TEM, and to ensure that only endocytic events were analyzed (Schmelz et al, 1994; Tooze et al, 1993). The TEM images showed notable qualitative differences in both macropinosome shape and virion-macropinosome association between VACV A28- and A28-ΔD8 samples (Fig. 3C). A28- virions were tightly wrapped causing the macropinosome to take on the shape of the virions within. Of note, the limiting membrane of macropinosomes were often found in contact with A28- virion tips. In contrast A28- ΔD8 containing macropinosomes remained circular with little evidence of virion-mediated membrane deformation. To analyze the curvature of the macropinosome membrane induced by single virions, Kappa, a Fiji plugin which measures curvature using B-splines, was used (Mary and Brouhard, 2019). Only virions in contact with the limiting macropinosome membrane, in a side-on orientation, were considered for analysis to avoid skewing the data with high curvature values around the tip of A28- virions. Using these parameters, a 1.3-fold drop in macropinosome membrane curvature around virions was seen in the absence of D8 (Fig. 3D).

These results are consistent with the model shown in Fig. 3e, whereby WT VACV virions bind to the cell surface in a side on orientation via A26-laminin, D8-chondroitin sulfate, and

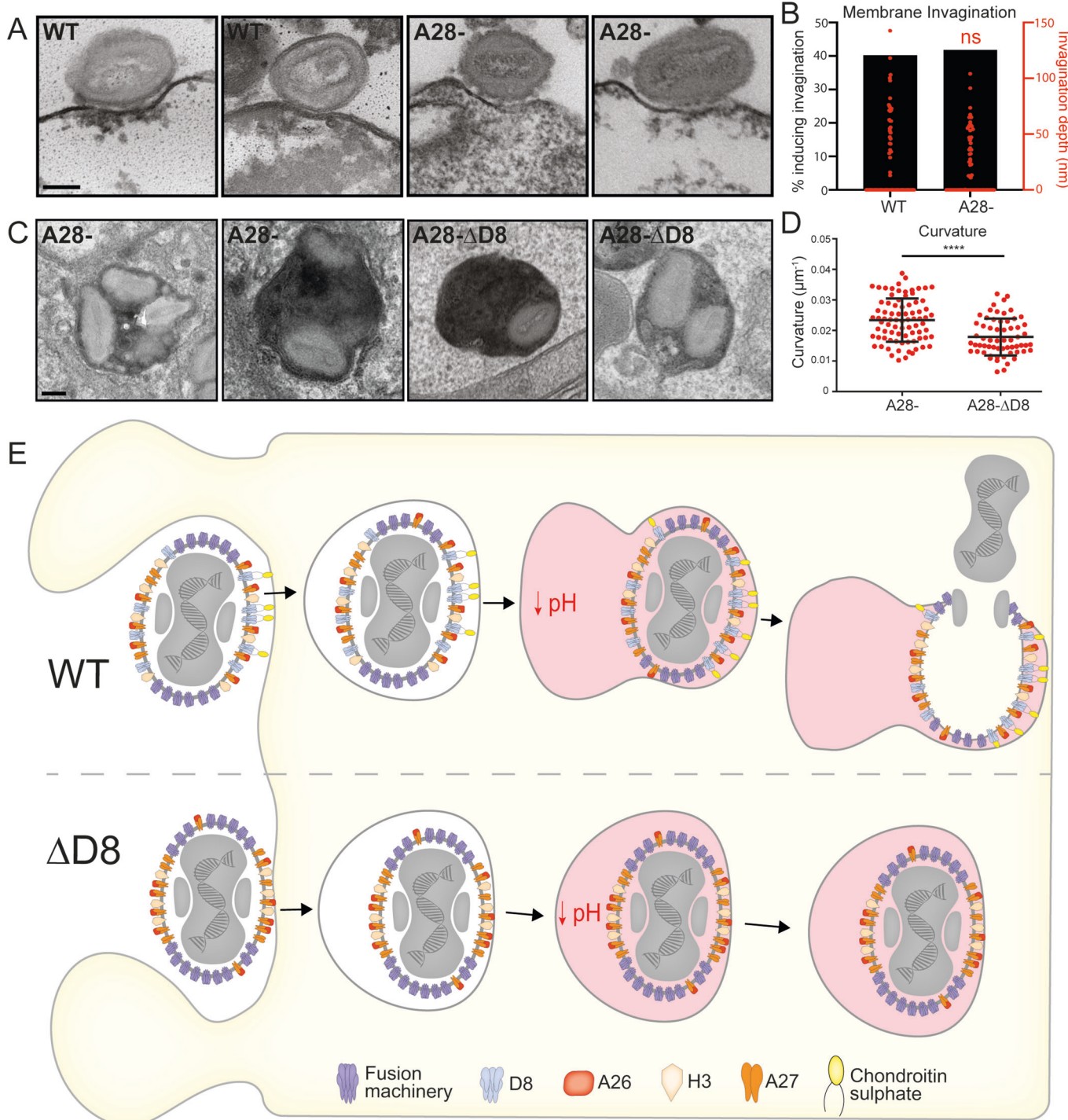

**Figure 3.** **D8-mediated invagination is a critical component of VACV low pH-dependent fusion.**

(**A**) Representative TEM images of WT and A28- virions bound to HeLa blebs at pH 5.0. Scale bar, 100 nm. (**B**) Quantification of WT and A28- virion % invagination and invagination depth comparing under pH 5.0 conditions (from (**A**)) ($n = 80$ virions/mutant). (**C**) Representative TEM images of A28- or A28-ΔD8 virions within macropinosomes. Scale bar, 100 nm. (**D**) Quantification of curvature of the macropinosome membrane around individual A28- or A28-ΔD8 virions (from (**F**)) using the Fiji plugin Kappa ($n > 60$ virions/mutant). (**E**) Model of low-pH mediated D8-mediated macropinosome membrane curvature and VACV fusion. Data information: Statistical analysis was performed using unpaired two-tailed $t$ tests (****$P < 0.0001$; ns, not significant ($P > 0.05$)). Source data are available online for this figure.

H3-heparan sulfate interactions before being macropinocytosed. As the macropinosome matures lowering the pH within, the fusion repressor A26 is removed, and D8-mediated macropinosome membrane curvature is triggered. This serves to bring the limiting membrane of the macropinosome in contact with the fusion machinery, located at virion tips, thereby increasing the likelihood of productive fusion. When virions lacking D8 (ΔD8) are macropinocytosed, their inability to drive membrane invagination slows the rate of VACV hemifusion fusion. Taken together our results strongly suggest that low pH dependent, D8-mediated membrane invagination is an important component of VACV fusion from macropinosomes

## VACV D8/CS interactions mediate virion binding and drive membrane curvature

Next, we turned to the cellular factors involved in this phenomenon. Like many viruses, VACV uses cellular glycosaminoglycans, such as heparan sulfate (HS) and chondroitin sulfate (CS), as cell surface attachment factors (Mercer et al, 2020, 2010b). As D8 is a well-defined CS-binding protein (Hsiao et al, 1999; Matho et al, 2014), it seemed likely that the D8-CS interaction was the driving force behind D8 membrane invaginating activity. To test this, we took advantage of WT L929 cells that display both HS and CS (HS+CS+), and the L929 derivatives gro2c which display only CS (HS-CS+), and sog9 which display neither HS or CS (HS-CS-) (Gruenheid et al, 1993; Banfield et al, 1995).

Confirming the importance of CS in VACV attachment, 39% less A4-EGFP VACV virions bound to cells lacking CS than to cells containing both HS and CS, or cells lacking HS alone (Figure EV3A). To investigate the role of CS in D8-mediated invagination we generated HS+CS+ and HS-CS- blebs—confirmed that VACV could bind to them (Figure EV3B)—and compared invagination under WT virions at pH 5.0. Membrane invagination under virions was evident in HS+CS+ blebs and absent in HS-CS- blebs (Fig. 4A). Quantification indicated that 39% of virions could induce invagination of HS+CS+ blebs, as opposed to just 8.5% in HS-CS- blebs, which was accompanied by a significant reduction in invagination depth (Fig. 4B). An invagination assay using ΔD8 virions on HS+CS+ blebs confirmed this was D8 driven (Figure EV3C). Consistent with the decreased rate of hemifusion observed with ΔD8 virus (Fig. 3B), the rate of VACV hemifusion on HS-CS- cells was reduced by 1.9-fold relative to HS+CS+ cells (Fig. 4C). Together with our binding experiments, these results indicate that CS—through its interaction with D8—is important for both virus attachment to the cell surface and the rate of low pH-dependent VACV fusion.

## VACV D8/CS-E interactions can compensate for the loss of H3/HS mediated binding

Although H3 binds HS (Lin et al, 2000), and our H3-binding experiments showed a significant reduction in VACV adsorption (Figure EV2C), we saw only a minor reduction in VACV binding to cells lacking HS (Figure EV3A). This was further corroborated with soluble GAG pre-incubation experiments. For this, A4-EGFP virions were pre-incubated with heparan, HS, CS-A, CS-E—which was reported to be the specific binding partner of D8 (Matho et al, 2014)—or laminin prior to adsorption onto HS+CS+ cells.

Analysis of VACV binding by flow cytometry indicated that pre-incubation with heparan, HS or CS-A had little impact on VACV attachment, while pre-incubation with CS-E or laminin blocked VACV binding by 30% (Figure EV3D). When CS-E pre-incubation experiments were repeated on HS-CS+ cells we found that CS-E reduced binding to HS-CS+ cells by a further 27% relative to HS+CS+ cells (Figure EV3E). Hence, CS-E is the predominant GAG used for VACV binding on HS+CS+ cells. That VACV binding to HS-CS+ cells relied on CS-E to an even greater extent suggested to us that the D8/CS-E interaction may be compensating for the loss of H3/HS interactions and thereby masking any HS-binding phenotypes.

As H3 and A27 are the two virion HS-binding proteins (Hsiao et al, 1998; Lin et al, 2000), and we saw no reduction in VACV adsorption with A27- virions (Figure EV2C), we subjected A27-virions to heparan, HS, CS-A, CS-E and laminin pre-incubation binding experiments to reveal any potential H3/HS binding phenotypes. In the absence of A27, pre-incubation with heparin and HS blocked VACV binding by 44 and 39%, respectively (Figure EV2C). Binding of A27- virions was also more sensitive to CS-E preincubation than WT virions (57% vs 30% reduction), and consistent with the absence of A26 from A27- VACV (Howard et al, 2008), laminin pre-incubation no longer reduced VACV binding. These results show that removal of A27 from VACV virions unmasks their H3/HS binding capacity and increases their reliance on D8/CS-E binding, which together is sufficient for A27-virions to overcome the loss of A26/laminin mediated binding.

To test if D8/CS-E interactions compensate for the loss of HS in the context of WT virions we compared the invagination depth of WT virions bound to HS+CS+ and HS-CS+ blebs at pH 5.0 using TEM. There was a significant increase in the depth of invagination under virions bound to HS-CS+ blebs compared to those bound to HS+CS+ blebs (Fig. 4D). Quantification showed that 12% more virions induced invagination on HS-CS+ cells and that the average depth of invagination was significantly increased (Fig. 4E). Consistent with a role for D8/CS-E mediated invagination in promoting hemifusion efficiency, R18-dequencing assays indicated that the rate of hemifusion was 2-fold higher on HS-CS+ cells than on HS+CS+ cells (Fig. 4F).

## VACV D8/CS-E interaction affinity is regulated by low pH

Having shown that D8 and CS-E are required for VACV low pH-dependent membrane invagination, we next asked if the interaction between D8 and CS-E showed any pH dependence. To confirm the interaction with CS-E, soluble D8 was incubated with biotinylated versions of CS-E or CS-A (0.5 and 5.0 μM) and subsequently immunoprecipitated using streptavidin coated beads. Immunoblot analysis for D8 showed, as previously reported (Matho et al, 2014) that D8 binds CS-E much more readily than CS-A (Fig. 4G). To test if D8/CS-E binding is pH-sensitive, D8/CS-E immunoprecipitation experiments we performed at neutral (7.4) and low (5.0) pH. Immunoblot analysis for D8 indicated that 26% more D8 bound to CS-E under pH 5.0 conditions (Fig. 4H). The increased affinity of D8 for CS-E at pH 5.0 suggested that D8 may undergo a conformational change at low pH. As D8 on the surface of intact MVs has been shown to be highly sensitive to the protease papain at pH 7.4 (Townsley and Moss, 2007), and papain's peptidase activity has been shown to be even more

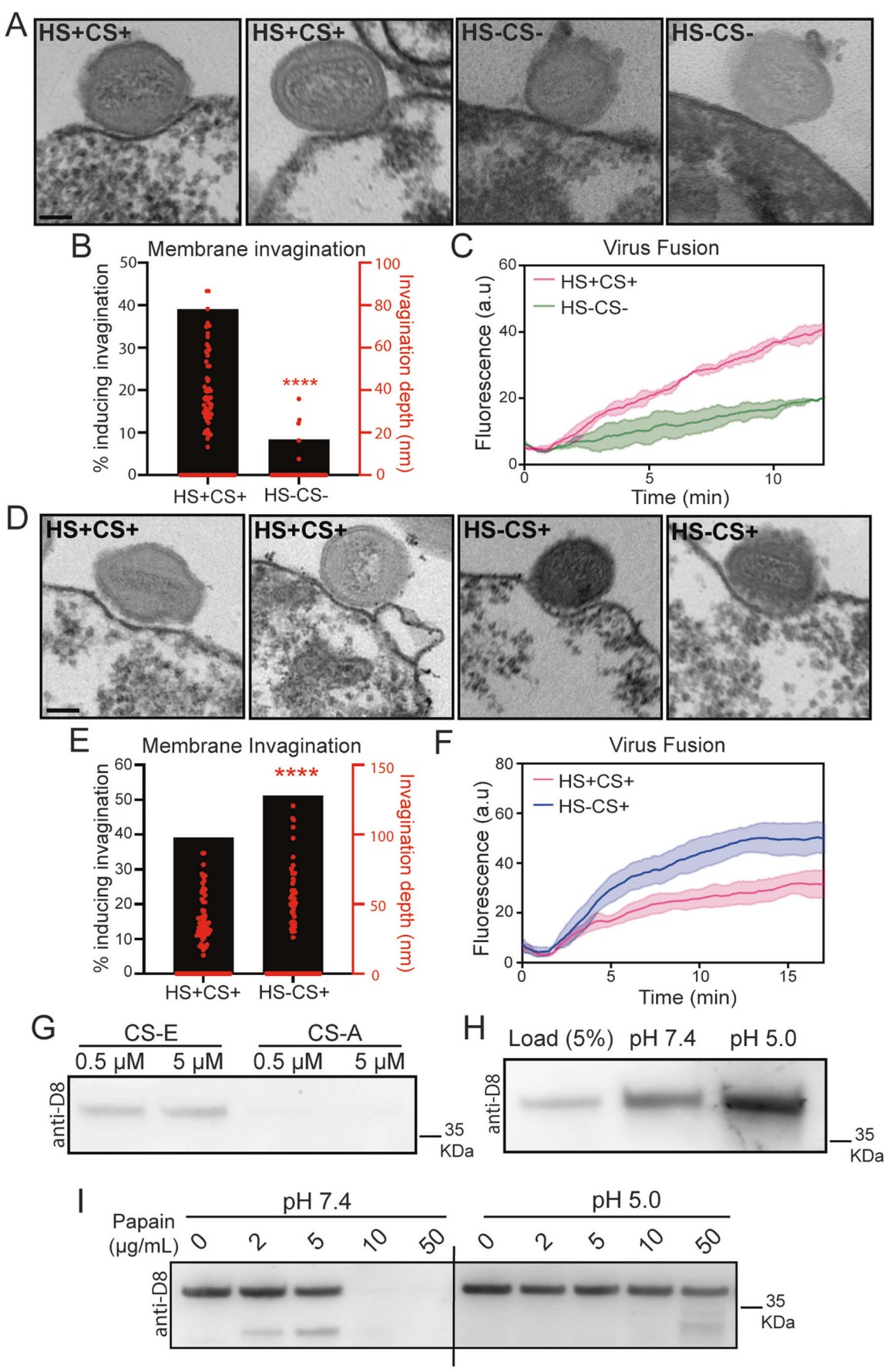

**Figure 4.   D8-mediated invagination requires pH sensitive chondroitin sulfate binding.**

(A) TEM images of WT VACV bound to HS + CS+ or HS-CS- cell-derived blebs at pH 5.0. Scale bar = 100 nm. (B) Quantification of % invagination and invagination depth of virions from A (n > 60 virions/mutant). (C) Hemifusion rates of WT virus on HS + CS+ and HS-CS- cells using the R18 dequenching assay. (D) TEM images of WT VACV bound to HS + CS+ or HS-CS+ cell-derived blebs at pH 5.0. Scale bars = 100 nm. (E) Quantification of % invagination and invagination depth of virions in (D) (n = 95 virions/mutant). (F) Hemifusion rates of WT virus on HS + CS+ and HS-CS+ cells using the R18 dequenching assay. (G) Immunoprecipitation analysis of D8 binding to biotinylated CS-E or CS-A. (H) Immunoprecipitation analysis of D8 binding to biotinylated CS-E at neutral (7.4) and low (5.0) pH. (I) Digestion of D8 from the surface of WT virions at neutral (7.4) and low (5.0) pH using varying concentrations of papain. Data information: (C, F) data are means ± SEM. G–I were performed in triplicate and representative blots shown. Statistical analysis was performed using unpaired two-tailed t tests (****P < 0.0001; ns, not significant (P > 0.05)). Source data are available online for this figure.

efficient at pH 5.0 than pH 7.0 (Hoover and Kokes, 1947), we employed a papain cleavage assay to investigate this. WT MVs were treated with increasing concentrations of papain (2–50 µg/ml) at pH 7.4 or pH 5.0. At pH 7.4, D8 underwent cleavage beginning at 2 µg/ml papain and was completely degraded at 50 µg/ml (Fig. 4I; left). In contrast, at pH 5.0 D8 remained stable in the presence of increasing papain, with minor cleavage occurring at 50 µg/ml (Fig. 4I, right). These results indicate that the affinity of D8 for CS-E is regulated by pH and suggest that this increased affinity is garnered by a low pH-dependent conformational change in D8 that makes it insensitive to papain.

Together these findings confirm CS-E as the cellular binding partner of D8 (Matho et al, 2014), define the importance of this interaction in assuring VACV low-pH dependent fusion efficiency, and uncover the built-in compensatory nature of VACV cell surface binding proteins.

## Discussion

Having shown that VACV binding and fusion machineries are organized into distinct functional domains that dictate virion cell binding and fusion orientation (Gray et al, 2019), we set out to characterize the relationship between side-on virion binding and tip-on virion fusion. To do this we established a cell-derived bleb model system (Biro et al, 2013). We show that blebs can be purified from multiple cell types and maintain the biologically relevant cell surface molecules needed to study virus binding and fusion.

Using blebs we determined that VACV side-on binding is static and that bound virions could undergo low-pH mediated hemifusion with blebs, i.e., virions do not change orientation to bring the virus fusion machinery into contact with the target membrane. This led to the discovery that VACV virions can induce target membrane invagination in the presence of low pH. We found that invagination is mediated by the binding protein D8 and its interaction with the GAG CS-E. We show that this phenomenon is required for efficient low pH dependent membrane fusion activity at the plasma membrane and within macropinosomes.

A search for functional domains suggests that D8 does not contain any features associated with membrane bending activity (McMahon and Gallop, 2005). No amphipathic helices or additional transmembrane domains that can be inserted into cellular membranes; nor any signaling domains that might serve to trigger the recruitment of cytosolic coat or cytoskeletal proteins were uncovered.

Interestingly, virus-induced membrane curvature has been described for SV40 in cells and GUVs and human norovirus on GUVs (Rydell et al, 2013; Ewers et al, 2010). For both, interactions between pentameric virus scaffolds and glycosphingolipids are

responsible for inducing plasma membrane curvature, and in the case of SV40 virus endocytosis (Ewers et al, 2010). Cholera toxin, which binds the ganglioside GM1 akin to SV40, has been shown to induce membrane curvature dependent upon multivalent binding and clustering of GM1 (Ewers et al, 2010; Maarouf Kabbani et al, 2020)

While the induction of membrane invagination seen here with VACV diverges in its pH dependence, it is similar in several ways. It is mediated by interactions between D8 and the GAG CS-E, which like GM1, is a component of the glycocalyx. EM structural studies suggest that D8 is a hexameric structure (Matho et al, 2014), which like SV40 pentamers provides multiple close-proximity binding sites (Kuo and Paszek, 2021). Consistent with the concept that membrane invagination driven by GM1 clustering requires multivalent binding, monomeric cholera toxin binding is not sufficient to induce membrane curvature (Ewers et al, 2010). Along these lines, hexameric D8 was shown to bind CS-E with greater avidity than monomeric D8 (Matho et al, 2014). Perhaps like SV40/GM1 mediated membrane bending, membrane invagination under VACV virions is driven by organized, multivalent D8-mediated clustering of CS-E.

Consistent with VACV entry by low pH-dependent endocytosis (Mercer and Helenius, 2008; Townsley and Moss, 2007; Huang et al, 2008; Mercer et al, 2010a), we found that the affinity of D8 for CS-E was greater at pH 5.0 than at pH 7.4. Thus, there is a correlation between low D8/CS-E affinity and VACV binding at pH 7.4, and high D8/CS-E affinity and membrane invagination at pH 5.0. As we have shown that D8-mediated membrane invagination facilitates virus fusion, it reasons that the different pH dependent affinities of D8 for CS-E provide a built-in mechanism to assure that VACV fusion does not occur at the plasma membrane but is delayed until virions reach late macropinosomes.

Consistent with this idea, there are now four low pH dependent processes identified that influence VACV entry: removal of the fusion suppressor A26 (Chang et al, 2012), protonation of the virus core (Schmidt et al, 2013), D8/CS-E mediated invagination and finally virus fusion. That removal of the fusion repressor (Townsley and Moss, 2007) and core activation can occur independent of target membrane contact suggests that these constitute the first low pH step, while D8/CS mediated invagination and virion fusion constitute the second. These findings are in line with previous studies showing that loss of D8 has a moderate impact on early virus production in single-step and 24-h virus yield assays (Niles and Seto, 1988; Hsiao et al, 1999).

Investigating hierarchy and redundancy between the VACV binding proteins using the four binding mutant VACVs in parallel, we found that A26 > D8 > H3 regarding virion cell surface attachment. These findings agree with VACV ligand receptor capture experiments, which found VACV A26 (the laminin binding

protein), α-dystroglycan (a laminin interacting protein) and CSPG4 (chondroitin sulfate proteoglycan 4) as potential VACV attachment factors in HeLa cells (Frei et al, 2012).

In this study, we found no direct role for A27 in virus binding. This is consistent with the finding that A27- virus shows no defect in 24 h virus yield (Hsiao et al, 1998; Vázquez et al, 1999; Gray et al, 2019). Soluble A27 has been shown to bind heparin in vitro, and soluble heparin to block soluble A27 binding to cells (Chung et al, 1998). Despite being assigned as a VACV HS-binding protein, no investigation of A27/HS cell surface binding in the context of intact VACV virions has been performed (Carter et al, 2005; Bengali et al, 2010, 2013; Whitbeck et al, 2009). Using A27- virions, we show that A27 is not required for VACV binding (Figure EV2C). Consistent with this, virion binding becomes significantly more sensitive to heparin, HS and CS-E preincubation in the absence of A27 (Figure EV3D vs. Fig. 4F). These results suggests that H3, the remaining heparin/HS binding protein (Lin et al, 2000), and D8 are more accessible on the virion surface in the absence of A27. While increased D8 accessibility correlates with the redistribution of D8 clusters on A27- MVs (Gray et al, 2019), that A27- virions show no binding defect suggests that D8 cluster redistribution does not impact virion binding. Consistent with the absence of A26 from A27- virions (Howard et al, 2008; Chang et al, 2013; Ching et al, 2009; Wang et al, 2014), we also found that A27- virions were no longer sensitive to laminin preincubation (Figure EV3D vs. Fig. 4F).

Collectively these results show that A27 is not a VACV binding protein, and that removal of A27 from VACV MVs unmasks H3 and D8. This in turn increases their respective HS and CS-E binding capacities, which is sufficient to overcome the loss of A26/laminin mediated binding seen with A27- virions. When considered in the context of our previous finding that A27 is required for EFC polarization, which in turn is needed for efficient fusion (Gray et al, 2019), we favor a model in which A27 indirectly regulates virion binding and fusion activities by acting as an organizer of MV membrane binding and fusion protein architecture. This likely explains why A27 has been attributed multiple, sometimes confounding, roles in VACV binding and fusion (Chang et al, 2019; Townsley and Moss, 2007; Rizopoulos et al, 2015; Vázquez et al, 1999).

These results together with our comparative studies on HS+CS+ and HS-CS+ cells, which indicate that VACV increased its usage of CS in the absence of HS, serve to highlight the degree to which VACV varies its use of HS and CS and laminin for cellular attachment depending on the experimental systems and cell types. These results suggests that VACV "adapts" attachment factor usage depending on which binding proteins and cell factors are available. This has also been seen with herpes simplex virus 1 (HSV-1), which adopts CS-E as a dominant attachment factor in the absence of HS (Bergefall et al, 2005). In both cases, this attachment factor switching phenomenon likely acts to expand the cell type promiscuity and host range of these viruses.

We have previously suggested that "the organization of the VACV membrane into functionally distinct domains has evolved as a mechanism to maximize virion binding and fusion efficiency for productive infection" (Gray et al, 2019). Here we demonstrate that VACV maximizes side-on binding and tip-directed virion fusion by directly linking these two processes through low pH dependent membrane invagination. In doing so, we further demonstrate that virion protein architecture is critical to virus function and extend the role of virus binding proteins beyond that of mere attachment factors.

## Methods

### Cells lines

BSC40 (kind gift of P. Traktman, Medical University of South Carolina, Charleston, SC, USA), human HeLa (ATCC, HeLa CCL-2), L929, Gro2c and Sog9 mouse cells (Zellbank, Friedrich-Loeffler-Institut) were cultivated in Dulbecco's modified Eagle's medium (DMEM; Gibco, Life Technologies 11320033) supplemented with 10% heat-inactivated fetal bovine serum (FBS; Life Technologies 10500064), 2 mM Glutamax (Life Technologies 35050038) and 1% pen-strep (pen-strep; Sigma P0781). All cell lines were mycoplasma free and tested for mycoplasma monthly.

### Viruses

Recombinant VACVs were generated in the Western Reserve (WR) strain. EGFP-A4(Schmidt et al, 2011), A27(+/−) EGFP-A4(Gray et al, 2019) and E-EGFP (Kilcher and Mercer, 2014) were previously described. ΔD8 and H3(-) were previously described as ΔD8vFire (Townsley and Moss, 2007) and vH3i (da Fonseca et al, 2000), respectively. ΔD8 EGFP-A4, H3(-) EGFP-A4 and WRΔA26 EGFP-A4 were constructed as previously described (Schmidt et al, 2013). To generate ΔA26 EGFP-A4, A26 was deleted at its endogenous locus with mCherry, for ΔA26ΔD8 EGFP-A4, A26 was replaced with lacZ ΔD8 EGFP-A4. For HA-H3 or A26-HA, the N- and C-terminus were HA-tagged respectively and recombinant virus selected for using E/L EGFP expression. Virus were selected and purified to homogeneity through four rounds of plaque purification and all resultant recombinant viruses confirmed by sequencing. All viruses were produced in BSC40 cells, and MVs band purified as previously described (Mercer and Helenius, 2008). A27(-) and H3(-) virus stocks were generated in the absence of isopropyl β-d-thiogalactopyranoside (IPTG; Sigma 16758). Plaque forming units/ml and particle counts—calculated from the optical density at 260 nm (Nichols et al, 2008)—were determined for each purified recombinant VACV MV stock. Viruses generated for this work are available upon request.

### Biosafety

All experiments performed with viruses were conducted in a class 2 biosafety approved laboratory, and in compliance with University of Birmingham, UK and Health and Safety Executive UK class 2 biosafety approved protocols.

### Mature virion (MV) yield

Confluent 6-wells of BSC40, HS+CS+, HS-CS+ or HS-CS- were infected at a multiplicity of infection (MOI) of 10 for 1 h, fed with full medium and infected cells collected at 24 hpi. Virus was released from cells by 3 rounds of freeze–thaw and plaque forming units/milliliter (pfu/ml) determined by serial dilution of BSC40 cell monolayers.

### Antibodies

Anti-L1 mouse monoclonal antibodies (clone 7D11) was provided by B. Moss (National Institute of Health) with permission of A.

Schmaljohn (University of Maryland). Anti-D8 rabbit polyclonal was provided by P. Traktman (Medical University of South Carolina). Anti-HA rabbit polyclonal antibodies were purchased from BioLegend (902302). Anti-A27 rabbit antibody (VMC39) was produced using purified recombinant baculovirus-expressed proteins by the Cohen lab (Aldaz-carroll et al, 2005).

## Bleb preparation

For bleb preparations, 3 confluent T75 flasks of cells were treated with 1.6 μM Latrunculin B (Sigma L5288) in DMEM for 15 min on an orbital shaker (700 rpm). The media was collected, detached cells pelleted (300 × g, 5 min) and the bleb-containing supernatant spun at 4000 × g to pellet blebs. The bleb pellet was resuspended in buffer (IB; 10 mM NaCl, 280 mM k-glutamic acid, 14 mM $Mg_2SO_4$, 13.34 mM $CaCl_2$, 5 mM Hepes) and filtered through a 5 μm filter (Cambridge Bioscience 43-10005-40). For ATP reconstitution, blebs were permeabilised with 50 mg/ml *Staphyloccus aureus* α-toxin (Hemolysin, Sigma H9395) and incubated with energy buffer (1 mM ATP, 1 mM UTP, 1 mM $MgCl_2$, 10 mM creatine phosphate (Sigma 2380), 1 mg/ml creatine phosphokinase (Sigma 2384)) for 20 min at room temperature (RT), before pelleting and resuspending.

## Binding orientation assay

Blebs were labeled with CellMask Deep Red Plasma Membrane Stain (Invitrogen C10045) and pelleted (300 × g for 10 min) onto fibronectin (Sigma F1141) coated CELLview glass bottom cell culture slides (Greiner Bio-One 43079). A4-EGFP WT or mutant viruses were R18 labeled and bound to blebs at 4 °C for 1 h. Samples were then washed and incubated with media adjusted with 100 mM 2-(*N*-morpholino)ethanesulfonic acid (MES) to pH 7.4 or pH 5.0 for 10 min at 37 °C. Samples were fixed with 4% formaldehyde 20 min, washed and imaged on an ELYRA PS.1 microscope (Zeiss).

## Structured illumination imaging

For viral membrane protein distribution, high-performance coverslips (18 × 18 mm, 1.5H, Zeiss) were washed and samples prepared and imaged as previously described (Gray et al, 2016).

## Polarity factor

Polarity factors were calculated as in Gray et al (2019). Briefly, polarity factors were calculated directly from the sets of particles that were used to generate the models (Fig. 2A). Following the crossvalidation particles were randomly divided into subsets of 50 particles and separately averaged. Radial profiles were generated from these images by transforming from $x$–$y$ coordinates to $r$–$\theta$. The radial profiles were divided into four regions according to the parameter $\varphi$ and the mean intensity within the viral membrane in these regions was determined.

## Flow cytometry

Cells were seeded to confluency in 96-well plates (Greiner-Bio One 650101) and incubated with equal numbers of particles of each recombinant EGFP-A4 VACV strain for 1 h at 4 °C. For cellular GAG and protein inhibition studies, virus was preincubated for 1 h at 4 °C with 100 μg/ml soluble H (Sigma H4784), HS (Sigma H7640), CS-A (Sigma C9819), laminin (Sigma L6274) or CS-E (Cosmo Bio CSR-NACS-E2(SQC)3), pelleted, washed and the virus- added to cells for 1 h at 4 °C. Unbound virus was removed by washing and cells fixed in 4% paraformaldehyde and harvested. Flow cytometry was performed on the Guava EasyCyte HT flow cytometer, recording the EGFP fluorescence with the 488 nm laser. Analysis of the flow cytometry data was performed with the GuavaSoft 3.3 analysis package (FlowJo).

## Electron microscopy

For virus bound to blebs, blebs were centrifuged (700 × g for 10 min) onto fibronectin (Sigma F1141) coated CELLview glass bottom cell culture slides (Greiner Bio-One 543079). Virus was bound to blebs at 4 °C for 1 h. Unbound virus was removed and samples incubated with media adjusted to either pH 7.4 or pH 5 with 100 mM MES for 10 min at 37 °C. Samples were fixed with 1.5% formaldehyde (TAAB F003) 2% glutaraldehyde (TAAB G011) for 20 min. For virus in macropinosomes, HeLa cells were grown to confluency on coverslips and recombinant virus bound at equal particle number at RT for 1 h. Unbound virus was removed and full medium supplemented with 10 mg/ml horseradish peroxidase (HRP). Samples were shifted to 37 °C for 1 h before fixation in 1.5% formaldehyde/2% glutaraldehyde in 0.1 M sodium cacodylate for 30 min. Samples were then incubated for 1 h in 1% osmium tetraoxide/1.5% potassium ferricyanide at 4 °C, treated with 1% tannic acid in 0.05 M sodium cacodylate for 45 min in the dark at RT, dehydrated in sequentially increasing concentrations of ethanol and embedded in Epon resin. The 70 nm thin sections were cut with a Diatome 45° diamond knife using an ultramicrotome (UC7, Leica). Sections were collected on 1 × 2 mm formvar-coated slot grids and stained with lead citrate. All samples were imaged using a transmission electron microscope (Tecnai T12, FEI) equipped with a charge-coupled device camera (SIS Morada, Olympus).

## TEM image analysis

For image analysis, invagination depth of bleb-bound virions was quantified using the Olympus SIS iTEM software. A straight line was drawn across the two edges of the invagination and the perpendicular distance to the inner leaflet of the bleb membrane was determined, signifying the "invagination depth." For macropinosome curvature analysis, the Kappa plugin (Mary and Brouhard, 2019) in Fiji (Schindelin et al, 2019) was used. In brief, an initialization curve was traced using a point-click method along the macropinosome membrane in contact with the virion membrane. This was then fit to the underlying data using an iterative minimization algorithm. The Bezier curve was extracted and the mean curvature along the entire curve reported.

## Early gene expression (RT-qPCR)

RT-qPCR was performed as described in Yakimovich et al (2017). Briefly, total RNA was extracted from infected HeLa cells using the RNeasy Plus Mini kit (Qiagen 74134) according to the manufacturer's instructions. Amplification of the VACV early gene, C11 and (GAPDH) cDNA was performed by qPCR (Mesa Blue qPCR

MasterMix Plus for SYBR assay; EurogentecB). Viral mRNA threshold cycle (CT) values are displayed as abundance normalized against GAPDH.

## Bulk fusion measurements

Bulk fusion was performed as described in Schmidt et al (2013). Briefly, Equal MV particles were labeled with R18 (R18, Thermo-Fisher Scientific O246) and bound to HeLa cells or blebs and the pH maintained at 7.4 or lowered to 5.0 by the addition of 100 mM MES.. After acquisition, total R18 was dequenched by the addition of 10% Triton X-100. R18 fluorescence was normalized to the signal intensity after the addition of Triton X-100. R18 fluorescence was measured using a Horiba FluoroMax-4 (Horiba Jobin Yvon) spectrofluorometer with an excitation wavelength of $560 \pm 5$ nm and an emission wavelength of $590 \pm 5$ nm.

## Immunoblot

Purified virions were heated at 95 °C for 10 min before separation on 4–12% Bis-Tris polyacrylamide gels (Thermo Fisher Scientific NW00125BOX) and transfer to nitrocellulose membranes. Membranes were blocked with 5% milk, 0.1% Tween-20 (TBST; Abcam ab64248) for 1 h at RT. Membranes were incubated with primary antibody D8 or L1 (1:1000) and bands visualized with (HRP)-coupled secondary antibody (1:5000; Cell Signalling, rabbit 7074S, mouse 7076S) on an ImageQuant LAS 4000 Mini (GE Life Sciences) with Luminata Forte Western HRP Substrate (Sigma WBLUF0500) for detection.

## Immunoprecipitation

In all, 5 μM purified D8 protein (MyBioSource) was incubated with 0.5 μM or 5 μM biotinylated versions of CS-E or CS-A (Creative PEGworks) for 3 h at RT. 0.5 mg of Dynabeads M280 streptavadin was added to each reaction and samples were incubated at RT for 3 h. Samples were then washed 4× with PBS and beads were resuspended in SDS PAGE buffer with 40 mM DTT for western blot analysis. Immunoprecipitations were carried out at pH 7.4 or 5.0 where indicated.

## Papain treatment assay

WT virions ($8 \times 10^7$ particles) were incubated at 37 °C for 3 min in pH 7.4 or 5.0 PBS adjusted with 100 mM MES. Increasing concentrations of papain, diluted in 5 mM L-cysteine at pH 7.4 or pH 5, were added to the virions and samples were incubated at 37 °C for 30 min. Papain was inactivated using 40 mM *N*-ethyl maleimide prior to pelleting and resuspension of virions was carried out with SDS PAGE buffer and 40 mM DTT for immunoblot analysis.

## Data availability

No primary datasets have been generated or deposited.

## Peer review information

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

## Acknowledgements

We thank B. Moss for providing the mutant viruses that were used in this study. Laura Pokorny is supported by the UCL Birkbeck MRC DTP. JJB is supported by MRC core funding to the MRC Laboratory for Molecular Cell Biology at University College London, award code (MC_U12266B). DA was funded by a Marie Skłodowska-Curie fellowship funded by the European Union (750673), RB was funded by the MRC Laboratory for Molecular Cell Biology PhD program. KEL and YM are supported by a Wellcome Trust Senior Research Fellowship (217191/Z/19/Z) to YM, PS, and TJK are supported by BBSRC grants (BB/P0098401 and BB/S017283/1) to TJK, and JM was supported by the European Research Council (649101-UbiProPox) and core funding to MRC Laboratory for Molecular Cell Biology (MC_UU_00012/7).

## Author contributions

**Laura Pokorny**: Conceptualization; Data curation; Formal analysis; Investigation; Methodology; Writing—original draft; Writing—review and editing. **Jemima J Burden**: Formal analysis; Investigation; Methodology; Writing—review and editing. **David Albrecht**: Formal analysis; Investigation; Methodology; Writing—review and editing. **Rebecca Bamford**: Formal analysis; Investigation; Writing—review and editing. **Kendra Leigh**: Formal analysis; Investigation; Writing—review and editing. **Pooja Sridhar**: Formal analysis; Investigation; Writing—review and editing. **Timothy J Knowles**: Formal analysis; Supervision; Investigation; Writing—review and editing. **Yorgo Modis**: Supervision; Methodology; Project administration; Writing—review and editing. **Jason Mercer**: Conceptualization; Supervision; Funding acquisition; Methodology; Writing—original draft; Project administration; Writing—review and editing.

## Disclosure and competing interests statement

The authors declare no competing interests.

# Expanded View Figure

**Figure EV1. Cell-derived membrane blebs as a minimal cell system.**

(A) To prepare blebs Latrunculin B is added to cells to induce blebbing. Cells are shaken to detach blebs, cellular debris removed by a slow spin step ($300 \times g$), blebs collected by a fast spin step ($4000 \times g$), and remaining large debris removed by filtration through a 5 µm pore filter. (B) Representative brightfield (LHS, scale bar; 1 µm) and TEM (RHS, scale bar; 500 nm) images of blebs after purification. (C) Histogram of bleb diameter range. (D) Blebs were scored for mono-, bi- and multi-lobulation ($n > 100$ blebs/repeat). (E) Blebs were stained for actin (green) and the plasma membrane (PM; magenta). Scale bar, 5 µm. (F) Bleb cortical actin was reconstituted (R) with ATP or not (NR), stained for both actin and the PM and the percentage of blebs with an actin cortex calculated ($n > 100$ blebs/repeat). (G) Stability of the actin cortex over time at 37 °C between NR and R blebs was determined by intensity measurements of the actin stain on z-projections ($n > 40$ blebs). Data information: For (D, F, G), data are means ± SD. Statistical analysis was performed using unpaired two-tailed *t* tests (ns, not significant ($P > 0.05$)).

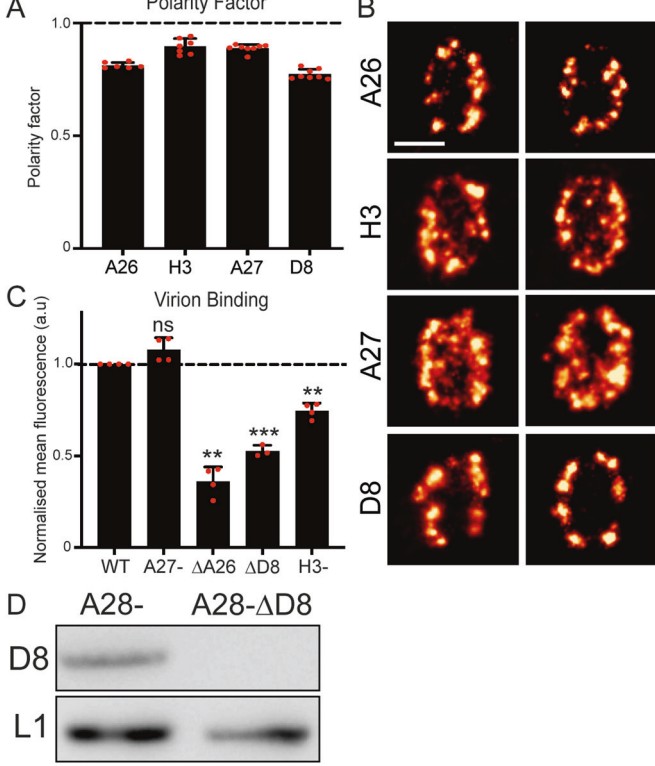

**Figure EV2. VACV binding proteins reside at virion sides and are differentially required for VACV binding.**

(A) Quantification of VACV binding protein polarity factors using data from the models in Fig. 2A. A polarity factor of less than one corresponds to concentration of the protein at the sides of MVs ($n = 50$ virions per repeat). (B) Representative STORM images of VACV binding proteins on individual MVs. Scale bar $= 200$ nm. (C) Binding affinities of WT and recombinant binding protein mutant VACVs on HS $+$ CS$+$ cells ($n = 4$ biological repeats). (D) Representative immunoblot of D8 protein packaging in A28- and A28-$\Delta$D8 virions. Molecular weight markers are indicated at right. Data information: (A, C) data are means $\pm$ SD. Statistical analysis was performed using unpaired two-tailed $t$ tests (***$P < 0.001$; **$P < 0.01$; ns, not significant ($P > 0.05$)).

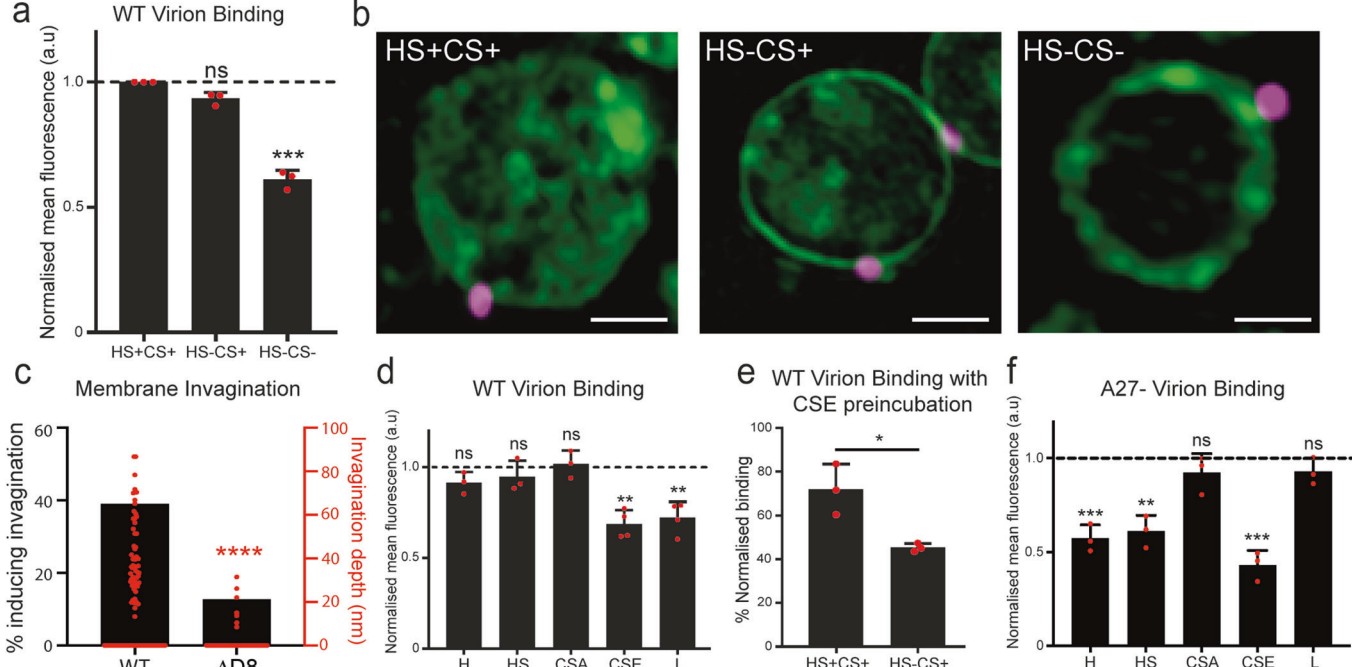

**Figure EV3.  CS-E is the major GAG used by VACV for binding.**

(A) Binding affinities of WT VACV on HS + CS + , HS-CS+ and HS-CS- cells. (B) SIM images of VACV bound to HS + CS + , HS-CS+ and HS-CS- derived blebs scale bar = 2 μm. A4-EGFP VACV (magenta) and PM (green). (C) Quantification of % invagination and invagination depth of WT and ΔD8 virions on HS + CS+ cells (n > 60 virions/mutant). (D) Binding affinities of WT VACV with GAG pre-incubation on HS + CS+ cells. (E) Binding affinities of WT virus with CS-E preincubation on HS + CS+ and HS-CS+ cells. Data is normalized to no CS-E preincubation on the given cell type. (F) Binding affinities of A27- virions with GAG pre-incubation on HS + CS+ cells. Data information: (A, D–F) data are means ± SD. Statistical analysis was performed using unpaired two-tailed t tests (***P < 0.001; **P < 0.01; *P < 0.05; ns, not significant (P > 0.05)).

