## [Peer Review File · EMBO Reports]

The vaccinia chondroitin sulfate binding protein drives host membrane curvature to facilitate fusion

Laura Pokorny, Jemima Burden, David Albrecht, Rebecca Bamford, Kendra Leigh, Pooja Sridhar, Timothy Knowles, Yorgo Modis, and Jason Mercer

DOI: [10.15252/embr.202357704](https://doi.org/10.15252/embr.202357704)

Corresponding author(s): Jason Mercer (j.p.mercer@bham.ac.uk)

Review Timeline:

Submission Date:	22nd Jun 23
Editorial Decision:	22nd Aug 23
Revision Received:	13th Nov 23
Editorial Decision:	6th Dec 23
Revision Received:	12th Dec 23
Accepted:	13th Dec 23

Editor: Achim Breiling

Transaction Report:

Dear Prof. Mercer,

Thank you for the submission of your manuscript to EMBO reports. I have received the reports from two of the three referees that were asked to evaluate your study, which can be found at the end of this message. A third referee had agreed to assess the manuscript, but has not delivered a report, despite several chaser from our side and promises by the referee to submit the report. In the interest of time, I have now decided to proceed with the submission.

As you will see, both referees think that these findings are of high interest. However, they have several comments, concerns, and suggestions to improve the manuscript, indicating that a major revision is necessary to allow publication of the study in EMBO reports. As the reports are below, and all the referee concerns need to be addressed, I will not detail them here.

Given the constructive referee comments, I would like to invite you to revise your manuscript with the understanding that all referee concerns must be addressed in the revised manuscript and in a detailed point-by-point response. Acceptance of your manuscript will depend on a positive outcome of a second round of review. It is EMBO reports policy to allow a single round of revision only and acceptance of the manuscript will therefore depend on the completeness of your responses included in the next, final version of the manuscript.

- 1) a .docx formatted version of the final manuscript text (including legends for main figures, EV figures and tables), but without the figures included. Figure legends should be compiled at the end of the manuscript text.
- 2) individual production quality figure files as .eps, .tif, .jpg (one file per figure), of main figures (up to 8) and EV figures (up to 5). Please upload these as separate, individual files upon re-submission.

- 4) a complete author checklist, which you can download from our author guidelines

(<https://www.embopress.org/page/journal/14693178/authorguide>). Please insert page numbers in the checklist to indicate where the requested information can be found in the manuscript. The completed author checklist will also be part of the RPF.

5) that primary datasets produced in this study (e.g. RNA-seq, CHIP-seq, structural and array data) are deposited in an appropriate public database. If no primary datasets have been deposited, please also state this in a dedicated section (e.g. 'No primary datasets have been generated and deposited'), see below.

The accession numbers and database should be listed in a formal "Data Availability" section (placed after Materials & Methods) that follows the model below. This is now mandatory (like the COI statement). Please note that the Data Availability Section is restricted to new primary data that are part of this study. This section is mandatory. As indicated above, if no primary datasets have been deposited, please state this in this section

Data availability

8) Regarding data quantification and statistics, please make sure that the number "n" for how many independent experiments were performed, their nature (biological versus technical replicates), the bars and error bars (e.g. SEM, SD) and the test used to calculate p-values is indicated in the respective figure legends (also for potential EV figures and all those in the final Appendix). Please also check that all the p-values are explained in the legend, and that these fit to those shown in the figure. Please provide statistical testing where applicable. Please avoid the phrase 'independent experiment', but clearly state if these were biological or technical replicates. Please also indicate (e.g. with n.s.) if testing was performed, but the differences are not significant. In case n=2, please show the data as separate datapoints without error bars and statistics. See also: <http://www.embopress.org/page/journal/14693178/authorguide#statisticalanalysis>

9) Please add scale bars of similar style and thickness to microscopic images, using clearly visible black or white bars (depending on the background). Please place these in the lower right corner of the images themselves. Please do not write on or near the bars in the image but define the size in the respective figure legend.

10) Please also note our reference format:

12) We now use CRediT to specify the contributions of each author in the journal submission system. CRediT replaces the author contribution section. Please use the free text box to provide more detailed descriptions and do not provide your final manuscript text file with an author contributions section. See also our guide to authors: <https://www.embopress.org/page/journal/14693178/authorguide#authorshipguidelines>

13) We would encourage you to use 'Structured Methods', our new Materials and Methods format. According to this format, the Materials and Methods section should include a Reagents and Tools Table (listing key reagents, experimental models, software and relevant equipment and including their sources and relevant identifiers) followed by a Methods and Protocols section in which we encourage the authors to describe their methods using a step-by-step protocol format with bullet points, to facilitate the adoption of the methodologies across labs. More information on how to adhere to this format as well as downloadable templates (.doc or .xls) for the Reagents and Tools Table can be found in our author guidelines (section 'Structured Methods'):

14) Please add up to 5 keywords to the manuscript text and order the manuscript sections like this, using these names: Title page - Abstract - Keywords - Introduction - Results - Discussion - Materials and Methods - Data availability section - Acknowledgements - Disclosure and Competing Interests Statement - References - Figure legends - Expanded View Figure legends

15) Please also add sub-titles to the results section to render the manuscript text more comprehensible.

16) Finally, please note that the abstract should not have more than 175 words.

I look forward to seeing a revised version of your manuscript when it is ready. Please let me know if you have questions or comments regarding the revision.

Yours sincerely,

Referee #1:

Pokorny et al. show that vaccinia virus binding proteins guide the orientation of virus binding and promote curvature of the host membrane towards entry fusion complex-containing virion tips to facilitate virus fusion. Using several assays, the authors provide evidence that D8, the chondroitin sulfate GAG binding protein, is involved in promoting membrane curvature following side-on virion binding. Using a cell-derived membrane bleb model system, the authors go on to show that the loss of D8 prevents virion-associated macropinosome membrane bending, disrupts fusion pore formation, and infection. The manuscript was clearly laid out and well-written. For the most part, the experimental design and generated data supported the conclusions that the authors came to. In this manuscript, the authors have uncovered a novel role for the poxvirus D8 viral protein, providing further evidence that the virus protein architecture is important for successful infection. I have included several comments/suggestions that will provide clarity to the manuscript and will help the authors to strengthen their submission.

Using a cell-derived membrane bleb system, the authors facilitate the quantification of large numbers of binding events in both fluorescence and electron microscopy-based assays of virus binding/hemi-fusion.

Major revisions

1. Figure 1 - the figure legend states that $n=150$, but only 3 data points for each treatment are displayed in 1d. Does each data point (in red) represent the mean of 150 individual virions per biological replicate? Or are you referring to the bar graph as the mean. Likewise, for 1f ($n=50$) and 1g ($n=100$), it appears that the authors plotted each virion invagination depth (red points), and then calculated the statistical significance on this measurement, correct? Please include the error bars for the mean of the biological triplicates for the "% inducing invagination" black bar graphs. Also, in the bottom panel of 1e (pH 5.0) do all these images represent side-on virions? If so, what does a tip-on virion look like when bound/invaginating a cell? The first panel in the bottom panel is difficult to distinguish the orientation of the dumbbell core structure to be certain that this is in side-on orientation.

2. In Figure 2, the authors conclude that an A27(-) mutant had no effect on the binding of VACV to membrane blebs (figure S2c). These data were somewhat surprising, given the importance that A27 has on regulating the protein architecture of VAC MV membranes (Gray et al. PMID 31285583) and its role in cell fusion. In the absence of A27, Gray et al showed that D8 clusters were re-distributed and suggest that A27 is required for polarized clustering of EFCs on MV tips. The authors should include some discussion of how their results here support the observations seen in Gray et al. In Figure 2f, the authors confirm through early gene expression that loss of D8 results in decreased kinetics of early gene expression. Including the impact of early gene

expression kinetics in the delA26 and H3(-) VAC would help to strengthen these data.

3. Using membrane blebs derived from cell lines lacking heparan sulfate GAGs and chondroitin sulfate GAGs, the authors next investigated the role of each GAG in membrane invagination. They observe that in the absence of HS and CS, they see a decrease in membrane invagination compared to when the GAGs are present. Interestingly, when HS is present, there seems to be a more pronounced membrane invagination, which the authors suggest may be an adaptation mechanism when the virus is binding to cells with differential levels of HS or CS. All of these studies were performed in Figure 4 with wt virus. It would be important to include the delD8 virus (and possibly the other binding mutants) in figure 4 as confirmation that the absence of the mutants in the L929 derivative cell lines are behaving as expected.

Minor revision

- The authors should carefully go through the manuscript as there were some minor typographical errors throughout (ie. p.1 second last line, 'questionable biologically relevance' should most likely be 'questionable biological relevance'; p2. Second paragraph, sentence starting "The pH 5.0... R18-dequeching..." should be "R18-dequenching").

Referee #2:

The question of how poxviruses such as vaccinia virus bind to cells and then mediate productive entry is complex and poorly understood. No high-affinity cellular receptor for poxvirus binding has been identified; rather, several (3) proteins on the virion surface that have affinity for glycoaminoglycans are thought to mediate the binding to the cell surface, and a 4th prevents fusion of the viral and cellular membranes until the correct signal has been given. Depending on the cell type and viral strain, entry is thought to occur either by direct fusion to the plasma membrane or via micropinocytosis followed by fusion-mediated release from the intracellular compartment into the cytosol. Fusion itself is mediated by a complex entry/fusion complex (EFC) composed of 11 viral proteins. This paper focuses on one of the GAG-binding proteins - D8 - which has been shown to bind to chondroitin sulfate (CS). The Mercer lab has previously used sophisticated imaging techniques to demonstrate that the EFC proteins are localized to the "tip" of the brick-shaped virions and the GAG-binding proteins (A27 and D8) to the sides of the virions.

Here, they use a nice model system using cell-derived membrane blebs, as well as intact cells, to study virion binding, membrane invagination and pH-dependent hemi-fusion. Key findings are: Virions generally bind to blebs on their side (3:1 side/tip ratio) and, in this orientation, mediate invagination of the cellular membrane when the pH is lowered from 7.4 to 5.0. When the GAG-binding proteins H3 or D8 are absent, the preferential binding orientation is lost. Importantly, when the D8 protein is absent and the process of virion binding and entry is assessed in intact HeLa cells (Fig 2), membrane invagination is greatly reduced in frequency, hemifusion is reduced (1.7-fold), and the post-entry onset of early gene expression is also reduced by 2.5-fold. In sum, these data demonstrate that D8 enhances virion binding, membrane invagination, and successful virion entry; although the effect may not be complete, the significance of the data is clear.

To further investigate the role of D8 during invagination, they generated a virus lacking A28 (a key component of the EFC) to prevent the progression past hemifusion of the viral and macropinosome membrane. Key findings are: EM analysis revealed a tight association between the virion membrane and the membrane of the macropinosome. In contrast, when both A28 and D8 were absent, the limiting membrane of the macropinosome appeared to be much less tightly associated with the virion membrane, and the overall shape of the organelle was far less deformed.

The authors then turn to question of whether CS, the ligand that binds to D8, played a key role in this process. Key findings are: Using WT L929 cells and derivatives that lack heparan sulfate (HS) or CS (Fig S4), they found that CS- cells show a reduction in the number of virions that bind (Fig S4). Pre-incubation of virions with CS-E (the subtype previously shown to be the best binding partner for D8), but not with CS-A or HS, reduced the subsequent binding of virions to cells. Pre-incubation of virions with CS-E prior to their analysis in the bleb-binding assay also reduced their ability to induce membrane invagination and hemifusion. In vitro binding assays revealed that the binding of D8 to CS-E is somewhat pH dependent, with increased binding seen at pH 5.0 (as compared to pH 7.4). Interestingly, D8 was more resistant to papain digestion at pH 5.0, suggesting that it may undergo a pH-dependent conformation that mediates this reduced protease-sensitivity (and perhaps enhance CS-E binding affinity).

Taken together, this is an innovative study that uses a variety of complementary approaches to further our understanding of the interaction of vaccinia virions with the host cell membrane. The authors have elucidated some key activities for the D8 protein - showing that virions bind primarily on their sides, that D8 is a prime mediator of cell binding, binds to CS-E but not CS-A and, most interestingly - mediates membrane invagination. This invagination seems to be a prerequisite for hemi-fusion of the virion and macropinosome membranes, and allows the EFC at the virion tip to function without the bound virions changing their polarity. D8 appears to bind the CS-E more robustly at low pH, which might have ramifications for low pH-dependent entry. The work is very well done, the paper is well written and the findings are provocative and will advance the field.

However, a few points do need to be addressed:

1. Regarding the pH-regulation of D8-CSE interaction: The authors present data showing that the binding of CS-E to D8 in their in vitro assay is regulated by pH (Fig 4H). However, virions bind to the intact cell surface at pH 7.4, and so the binding to CS at the plasma membrane MUST be sufficient if D8 is a major contributor to cell binding. How might enhanced D8:CS-E interaction in the mature macropinosome be relevant to the initial cell binding? The authors MUST discuss this further and provide a more meaningful discussion of how and why the pH regulation of D8:CS-E binding might be relevant in vivo?
2. Regarding the reduced papain-sensitivity of D8 at pH 5.0: The authors show (Fig 4i) that purified D8 seems to show a reduced sensitivity to papain at low pH, and argue that this might reflect a pH-induced conformational change that would be relevant to the milieu of the mature macropinosome. Have they addressed the possibility that papain is less active under the conditions of their assay at low pH? They should monitor a CONTROL substrate in order to rule out - or lend credence to - this technical possibility.
3. Regarding the differential reliance of virions on CS to bind cells depending on whether HS is present or absent. On p. 11, the authors state that "VACV increased its usage of CS in the absence of HS" and suggest that viruses may "switch" their GAG preference depending on what's available. Considering that the authors haven't provided any quantification of GAG levels on the cell surface (or in blebs, for that matter), might it not be that the levels of CS-E on HS- cells are higher than they are on HS+ cells? Might there be a competition for cell surface occupancy? The authors' interpretation seems overly assertive here, given the lack of much data. At a minimum, the interpretation suggested here (and any others) should be included.
4. Regarding the magnitude of the effects seen in this study. This report presents cleanly executed and well-quantitated experiments that provide a coherent story that is "qualitatively" compelling. Still, although the significance (both statistical and biological) of the findings is strong, the order of magnitude of the effects seen is in the range of 1.5 - 7 fold. Is this what the authors would expect from these types of studies? How do their results on D8 virions compare with the published work on the phenotype of virions lacking D8 or encoding temperature-sensitive mutants? It is important to explain whether the somewhat modest effects are a function of the redundancy of the GAG-binding proteins - which doesn't seem entirely likely - or the type of assays used.
5. Regarding the demonstration that the D8 protein plays a key role in inducing membrane invagination at low pH after virion binding. Perhaps the most intriguing finding of this study is the clear demonstration that bound virions containing D8 induce membrane invagination at low pH. This finding has significant implications for the process of subsequent membrane fusion and virion entry into the cytoplasm. The authors do not speculate - at all - about how this might occur. Does D8 have any motifs associated with membrane bending? Does D8 associate with any cellular proteins that are known to deform membranes and facilitate invagination? Some discussion of the possibilities should be provided.

Referee #1:

Pokorny et al. show that vaccinia virus binding proteins guide the orientation of virus binding and promote curvature of the host membrane towards entry fusion complex-containing virion tips to facilitate virus fusion. Using several assays, the authors provide evidence that D8, the chondroitin sulfate GAG binding protein, is involved in promoting membrane curvature following side-on virion binding. Using a cell-derived membrane bleb model system, the authors go on to show that the loss of D8 prevents virion-associated macropinosome membrane bending, disrupts fusion pore formation, and infection. The manuscript was clearly laid out and well-written. For the most part, the experimental design and generated data supported the conclusions that the authors came to. In this manuscript, the authors have uncovered a novel role for the poxvirus D8 viral protein, providing further evidence that the virus protein architecture is important for successful infection. I have included several comments/suggestions that will provide clarity to the manuscript and will help the authors to strengthen their submission.

Using a cell-derived membrane bleb system, the authors facilitate the quantification of large numbers of binding events in both fluorescence and electron microscopy-based assays of virus binding/hemi-fusion.

Major revisions

1. Figure 1 - the figure legend states that $n=150$, but only 3 data points for each treatment are displayed in 1d. Does each data point (in red) represent the mean of 150 individual virions per biological replicate? Or are you referring to the bar graph as the mean. Likewise, for 1f ($n=50$) and 1g ($n=100$), it appears that the authors plotted each virion invagination depth (red points), and then calculated the statistical significance on this measurement, correct? Please include the error bars for the mean of the biological triplicates for the "% inducing invagination" black bar graphs.

We apologize for the lack of clarity. For figure panel 1d each of the red data points represents the ratio of virions bound side on vs. tip on for $n=50$ virions/replicate. Three biological replicates were performed resulting in ($n=150$ virions). The black bars represent the mean of the three biological replicates. We have corrected the figure legend to read: "For **d** ($n=50$ virions/replicate), **f** ($n=50-70$ virions/replicate) and **g** ($n=50-70$ virions/replicate) data are mean \pm standard deviation (SD) of biological triplicates."

For figure panels 1f and 1g we have now added the error bars for the mean of the biological triplicates, as requested.

Also, in the bottom panel of 1e (pH 5.0) do all these images represent side-on virions? If so, what does a tip-on virion look like when bound/invaginating a cell? The first panel in the bottom panel is difficult to distinguish the orientation of the dumbbell core structure to be certain that this is in side-on orientation.

For figure panel 1e we have replaced the micrograph with one in which the dumbbell shape of the core is clearly visible. We have previously published the binding and fusion tip/side

orientation data for VACV virions on cells (gray et al Nat Microbiol. 2019 Oct;4(10):1636-1644). We found that 98% of virion binding events were side on. We have included some images of tip-bound invaginating virions here for the reviewer. Tip bound virions on rare occasion cause shallow invaginations (a). When more deeply invaginated tip-bound virions were observed there was always a degree of virion side contact with the plasma membrane, which we suspect is driven by D8 (b,c).

TEM images of WT VACV bound to HeLa cell derived blebs at pH 5.0. Examples of virion contact with the bleb membrane occurring at the tips of the virions.

2. In Figure 2, the authors conclude that an A27(-) mutant had no effect on the binding of VACV to membrane blebs (figure S2c). These data were somewhat surprising, given the importance that A27 has on regulating the protein architecture of VACV MV membranes (Gray et al. PMID 31285583) and its role in cell fusion. In the absence of A27, Gray et al showed that D8 clusters were re-distributed and suggest that A27 is required for polarized clustering of EFCs on MV tips. The authors should include some discussion of how their results here support the observations seen in Gray et al.

We thank the reviewer for this suggestion. We have now added a GAG-preincubation experiment with A27- virions (see Expanded View Fig. 4f) which was originally performed with the WT virion GAG-preincubation experiment (Expanded View Fig. 4d). We did not include this data originally to retain focus on D8, but now require it to support further discussion on the role of A27 in VACV virion architecture.

In this study, we found no direct role for A27 in virus binding. Soluble A27 has been shown to bind heparin *in vitro*, and soluble heparin to block soluble A27 binding to cells^{1,2}. Despite being assigned as a VACV HS-binding protein, no investigation of A27/HS cell surface binding in the context of intact VACV virions has been performed. Using A27- virions we show that A27 is not required for VACV binding (Expanded View Fig. 2c). Consistent with this, virion binding becomes significantly more sensitive to heparin, HS and CS-E preincubation in the absence of A27 (Expanded View Fig. 4d vs. 4f). These results suggests that H3, the remaining heparin/HS binding protein³, and D8 are more accessible on the virion surface in the absence of A27. While increased D8 accessibility correlates with the redistribution of D8 clusters on A27- MVs⁴, that A27- virions show no binding defect suggests that D8 cluster redistribution does not impact virion binding. Consistent with the absence of A26 from A27- virions⁵⁻⁸, we also found that A27- virions were no longer sensitive to laminin preincubation (Expanded View Fig. 4d vs. 4f).

Collectively these results show that A27 is not a VACV binding protein, and that removal of A27 from VACV MVs unmasks H3 and D8. This in turn increases their respective HS and CS-E binding capacities, which is sufficient to overcome the loss of A26/laminin mediated binding seen with A27- virions. When considered in the context of our previous finding that A27 is required for EFC polarization, which in turn is needed for efficient fusion⁴, we favor a model in which A27 indirectly regulates virion binding and fusion activities by acting as an organizer of MV membrane binding and fusion protein architecture. This likely explains why A27 has been attributed multiple, sometimes confounding, roles in VACV binding and fusion⁹⁻¹².

We have added this explanation to the discussion section of the manuscript.

In Figure 2f, the authors confirm through early gene expression that loss of D8 results in decreased kinetics of early gene expression. Including the impact of early gene expression kinetics in the Δ A26 and H3(-) VAC would help to strengthen these data.

We thank the reviewer for the suggestion. We have now included H3- in the R18 hemi-fusion experiment (Fig. 2e), as fusion is a more direct upstream measurement of the impact of D8 loss on virus entry. We also have added early gene expression data in for H3- in Fig. 2f. The H3- virions act very similar to WT virions showing no defect in fusion kinetics (see new Figure 2e and Fig. 2f).

Figure for referee with unpublished data and its description has been removed upon request by the authors.

3. Using membrane blebs derived from cell lines lacking heparan sulfate GAGs and chondroitin sulfate GAGs, the authors next investigated the role of each GAG in membrane invagination. They observe that in the absence of HS and CS, they see a decrease in membrane invagination compared to when the GAGs are present. Interestingly, when HS is present, there seems to be a more pronounced membrane invagination, which the authors suggest may be an adaptation mechanism when the virus is binding to cells with differential levels of HS or CS. All of these studies were performed in Figure 4 with wt virus. It would be important to include the delD8 virus (and possibly the other binding mutants) in figure 4 as

confirmation that the absence of the mutants in the L929 derivative cell lines are behaving as expected.

With respect, Fig. 4 was designed to assess the role of CS-E/D8 interactions in overcoming any HS requirement during the binding of WT virions. Inclusion of VACV mutants here would not control for the behavior of the cell lines, but rather test for compensatory binding interaction(s) between GAGs and the various VACV binding proteins. While this could be interesting based on the findings of the new Expanded View Fig. 4f, these experiments are beyond the scope of this manuscript.

Expanded View Fig. 4a and 4d show that the WT virion binding profile of HS+CS+, HS-CS+ and HS-CS- cell lines mirrors that seen in GAG-preincubation experiments in HeLa cells. The lack of, or preincubation of virions with, HS has no impact on virion binding to either cell type, while the lack of, or preincubation of virions with, CS results in reduced virus binding in both cell types. In addition, WT virion binding to HS-CS+ remains sensitive to CS-E preincubation as expected (Expanded View Fig. 4e). Based on these binding phenotypes we conclude that the HS+CS+, HS-CS+ and HS-CS- cell lines are acting as expected.

As D8 is the only CS binding protein on virions, we have now included a low pH-invagination assay on HS+CS+ cells comparing WT and Δ D8 VACV to assure that the Δ D8 virus retains its defective invagination phenotype in this cell type. This data is now included as (Expanded View Fig. 4c)

Minor revision

- The authors should carefully go through the manuscript as there were some minor typographical errors throughout (ie. p.1 second last line, 'questionable biologically relevance' should most likely be 'questionable biological relevance'; p2. Second paragraph, sentence starting "The pH 5.0... R18-dequeching..." should be "R18-dequenching").

We thank the reviewer for pointing these out, we have carefully edited the manuscript for typographical errors

Referee #2:

The question of how poxviruses such as vaccinia virus bind to cells and then mediate productive entry is complex and poorly understood. No high-affinity cellular receptor for poxvirus binding has been identified; rather, several (3) proteins on the virion surface that have affinity for glycoaminoglycans are thought to mediate the binding to the cell surface, and a 4th prevents fusion of the viral and cellular membranes until the correct signal has been given. Depending on the cell type and viral strain, entry is thought to occur either by direct fusion to the plasma membrane or via micropinocytosis followed by fusion-mediated release from the intracellular compartment into the cytosol. Fusion itself is mediated by a

complex entry/fusion complex (EFC) composed of 11 viral proteins. This paper focuses on one of the GAG-binding proteins - D8 - which has been shown to bind to chondroitin sulfate (CS). The Mercer lab has previously used sophisticated imaging techniques to demonstrate that the EFC proteins are localized to the "tip" of the brick-shaped virions and the GAG-binding proteins (A27 and D8) to the sides of the virions.

Here, they use a nice model system using cell-derived membrane blebs, as well as intact cells, to study virion binding, membrane invagination and pH-dependent hemi-fusion. Key findings are: Virions generally bind to blebs on their side (3:1 side/tip ratio) and, in this orientation, mediate invagination of the cellular membrane when the pH is lowered from 7.4 to 5.0. When the GAG-binding proteins H3 or D8 are absent, the preferential binding orientation is lost. Importantly, when the D8 protein is absent and the process of virion binding and entry is assessed in intact HeLa cells (Fig 2), membrane invagination is greatly reduced in frequency, hemifusion is reduced (1.7-fold), and the post-entry onset of early gene expression is also reduced by 2.5-fold. In sum, these data demonstrate that D8 enhances virion binding, membrane invagination, and successful virion entry; although the effect may not be complete, the significance of the data is clear.

To further investigate the role of D8 during invagination, they generated a virus lacking A28 (a key component of the EFC) to prevent the progression past hemifusion of the viral and macropinosome membrane. Key findings are: EM analysis revealed a tight association between the virion membrane and the membrane of the macropinosome. In contrast, when both A28 and D8 were absent, the limiting membrane of the macropinosome appeared to be much less tightly associated with the virion membrane, and the overall shape of the organelle was far less deformed.

The authors then turn to question of whether CS, the ligand that binds to D8, played a key role in this process. Key findings are: Using WT L929 cells and derivatives that lack heparan sulfate (HS) or CS (Fig S4), they found that CS- cells show a reduction in the number of virions that bind (Fig S4). Pre-incubation of virions with CS-E (the subtype previously shown to be the best binding partner for D8), but not with CS-A or HS, reduced the subsequent binding of virions to cells. Pre-incubation of virions with CS-E prior to their analysis in the bleb-binding assay also reduced their ability to induce membrane invagination and hemifusion. In vitro binding assays revealed that the binding of D8 to CS-E is somewhat pH dependent, with increased binding seen at pH 5.0 (as compared to pH 7.4). Interestingly, D8 was more resistant to papain digestion at pH 5.0, suggesting that it may undergo a pH-dependent conformation that mediates this reduced protease-sensitivity (and perhaps enhance CS-E binding affinity).

Taken together, this is an innovative study that uses a variety of complementary approaches to further our understanding of the interaction of vaccinia virions with the host cell membrane. The authors have elucidated some key activities for the D8 protein - showing that virions bind primarily on their sides, that D8 is a prime mediator of cell binding, binds to CS-E but not CS-A and, most interestingly - mediates membrane invagination. This invagination seems to be a prerequisite for hemi-fusion of the virion and macropinosome membranes, and allows the EFC at the virion tip to function without the bound virions changing their polarity. D8 appears to bind the CS-E more robustly at low pH, which might

have ramifications for low pH-dependent entry. The work is very well done, the paper is well written and the findings are provocative and will advance the field.

However, a few points do need to be addressed:

1. Regarding the pH-regulation of D8-CSE interaction: The authors present data showing that the binding of CS-E to D8 in their in vitro assay is regulated by pH (Fig 4H). However, virions bind to the intact cell surface at pH 7.4, and so the binding to CS at the plasma membrane MUST be sufficient if D8 is a major contributor to cell binding. How might enhanced D8:CS-E interaction in the mature macropinosome be relevant to the initial cell binding? The authors MUST discuss this further and provide a more meaningful discussion of how and why the pH regulation of D8:CS-E binding might be relevant in vivo?

We agree with the reviewer, the interaction between D8 and CS-E at pH 7.4 is sufficient for virus binding as illustrated in Expanded View Fig. 2a.

We have now added the following to the discussion: "Consistent with VACV entry by low pH-dependent endocytosis^{11,14-16} we found that the affinity of D8 for CS-E was greater at pH 5.0 than at pH 7.4. Thus, there is a correlation between low D8/CS-E affinity and VACV binding at pH 7.4, and high D8/CS-E affinity and membrane invagination at pH 5.0. As we have shown that D8-mediated membrane invagination facilitates virus fusion, it reasons that the different pH dependent affinities of D8 for CS-E provide a built-in mechanism to assure that VACV fusion does not occur at the plasma membrane but is delayed until virions reach late macropinosomes.

Consistent with this idea, there are now four low pH dependent processes identified that influence VACV entry:...."

2. Regarding the reduced papain-sensitivity of D8 at pH 5.0: The authors show (Fig 4i) that purified D8 seems to show a reduced sensitivity to papain at low pH, and argue that this might reflect a pH-induced conformational change that would be relevant to the milieu of the mature macropinosome. Have they addressed the possibility that papain is less active under the conditions of their assay at low pH? They should monitor a CONTROL substrate in order to rule out - or lend credence to - this technical possibility.

The experiment shown in Fig. 4i was performed on intact WT virions not purified D8. As there is no existing data on papain cleavage of VACV proteins in the context of virions at pH 7.4 vs. pH 5.0 there are no viral proteins that can serve as an appropriate control.

However, it has been demonstrated that the peptidase activity of papain is optimal at pH 5.0, being half as active at pH 7.0. That we observe reduced papain sensitivity of D8 at pH 5.0 vs. pH 7.4, would suggest that the difference in papain sensitivity is due to a change in D8 on the virion surface as opposed to reduced papain activity.

We have now added this information and reference to the manuscript.

3. Regarding the differential reliance of virions on CS to bind cells depending on whether HS is present or absent. On p. 11, the authors state that "VACV increased its usage of CS in the

absence of HS" and suggest that viruses may "switch" their GAG preference depending on what's available. Considering that the authors haven't provided any quantification of GAG levels on the cell surface (or in blebs, for that matter), might it not be that the levels of CS-E on HS- cells are higher than they are on HS+ cells? Might there be a competition for cell surface occupancy? The authors' interpretation seems overly assertive here, given the lack of much data. At a minimum, the interpretation suggested here (and any others) should be included.

We thank the reviewer for this comment, we had also considered this possibility. However, the GAG pre-incubation experiments on WT virions (Expanded View Fig. 4d and e) and A27- virions (now included as Expanded View Fig. 4f, see response to reviewer 1, point 2) suggest that there is indeed a "switch" in GAG usage depending on viral and cellular binding factors available.

Regarding the CS-E pre-incubation experiment (Expanded View Fig. 4e), if HS-CS+ cells expressed more CS-E than HS+CS+ cells, it reasons that there would be more CS-E available on the HS-CS+ cell surface for virus binding. This in turn should make WT virions less sensitive to CS-E preincubation blocking on HS-CS+ than on HS+CS+ cells. However, we see the opposite, suggesting that in the absence of HS, WT virions "switch" GAG usage becoming more reliant on CS-E for binding.

This is further supported by GAG pre-incubation experiments on A27- virions (Expanded View Fig 4f). We subjected WT and A27- virions to heparan, HS, CS-A, CS-E and laminin pre-incubation binding experiments. We found that A27- virion binding is significantly more sensitive to heparin, HS and CS-E preincubation in the absence of A27 (Expanded View Fig. 4d vs. 4f). Consistent with the absence of A26 from A27- virions⁵⁻⁸, we also found that A27- virions were no longer sensitive to laminin preincubation (Expanded View Fig. 4d vs. 4f).

As A27- virions display no binding defect (Expanded View Fig. 2c) these results suggests that the heparin/HS binding protein H3³, and D8 are more accessible on the virion surface in the absence of A27. This in turn increases their respective HS and CS-E binding capacities, which is sufficient to overcome the loss of A26/laminin mediated binding seen with A27- virions. Put simply, WT virions switch their binding requirements from A26/laminin to H3/Heparin/HS and increase D8/CS-E requirements in the absence of A27.

4. Regarding the magnitude of the effects seen in this study. This report presents cleanly executed and well-quantitated experiments that provide a coherent story that is "qualitatively" compelling. Still, although the significance (both statistical and biological) of the findings is strong, the order of magnitude of the effects seen is in the range of 1.5 - 7 fold. Is this what the authors would expect from these types of studies? It is important to explain whether the somewhat modest effects are a function of the redundancy of the GAG-binding proteins - which doesn't seem entirely likely - or the type of assays used.

Regarding the magnitude of the effects and redundancy, this is exactly what we would expect based on the use of multiple viral binding protein-cellular attachment factor combinations reported here and on our previous work on the VACV EFC⁴.

For binding, by directly comparing single binding protein deletion mutants, we show that three VACV proteins (A26, D8, and H3) contribute to virus binding to different extents and that the loss of individual proteins does not completely eliminate virus binding (Expanded View Fig. 2c). We further show that cellular attachment factor usage and reliance switches based on viral binding factors available (Expanded View Fig. 4d vs. f). This strongly suggested that the VACV binding proteins are not redundant but act in a compensatory manner.

For D8-mediated, pH-dependent membrane curvature, we don't believe it is absolutely required for virus entry but as stated in the title, it facilitates virus fusion. This is reflected in the delayed fusion and gene expression kinetics observed in the absence of D8. The magnitude of effect on fusion seen in the absence of D8 is very similar to what we observed in the absence of EFC polarization⁴. These results are consistent with the importance of virus architecture for assuring efficient virus fusion.

How do their results on Δ D8 virions compare with the published work on the phenotype of virions lacking D8 or encoding temperature-sensitive mutants?

As far as we are aware there are no *ts*D8 viruses. However, the initial characterization of D8 was performed by transcriptional mapping, antibody production and the construction of a mutant D8 virus by frameshift¹⁷. The mutant D8 virus lacked the C-terminal 56 amino acids of D8. As the mutation eliminated the trans-membrane domain, and the authors had a difficult time detecting the protein in virion fractionation, it was suggested that the mutant virus was no longer on the surface on mature virions. Single-step growth curves were used to assess the impact on virus yield. While the authors concluded there was no defect, a 0.5 log loss in yield is observed in the mutant virus at the earliest time point assayed (6hpi). This result is consistent with the early binding and fusion kinetics defects we observe in the absence of D8, and similar to the early fusion kinetics defects we have described upon disruption of EFC polarization⁴.

Consistent with our findings, Hsiao *et al.* showed that soluble D8 competes for VACV binding, that CS competes with soluble D8 binding, that soluble D8 does not bind to HS-CS-cells, and that a D8- VACV displays a 1-log defect in 24-hour virus yield. Consistent with our A27- VACV results, the Hsiao study also showed that the loss of A27 from D8- virions resulted in no additional decrease in yield.

We have now added references to these papers to the discussion section of the manuscript.

5. Regarding the demonstration that the D8 protein plays a key role in inducing membrane invagination at low pH after virion binding. Perhaps the most intriguing finding of this study is the clear demonstration that bound virions containing D8 induce membrane invagination at low pH. This finding has significant implications for the process of subsequent membrane fusion and virion entry into the cytoplasm. The authors do not speculate - at all - about how this might occur. Does D8 have any motifs associated with membrane bending? Does D8 associate with any cellular proteins that are known to deform membranes and facilitate invagination? Some discussion of the possibilities should be provided.

Functional domain searching suggests that D8 does not contain any features associated with membrane bending activity¹⁸. No amphipathic helices or transmembrane domains that can be inserted into cellular membranes; nor any signaling domains that might serve to trigger the recruitment of cytosolic coat or cytoskeletal proteins were uncovered.

We have added to the discussion a section on how D8 might facilitate membrane invagination.

Bibliography

1. Chung, C. S., Hsiao, J. C., Chang, Y. S. & Chang, W. A27L protein mediates vaccinia virus interaction with cell surface heparan sulfate. *J Virol* **72**, 1577–1585 (1998).
2. Hsiao, J. C., Chung, C. S. & Chang, W. Cell surface proteoglycans are necessary for A27L protein-mediated cell fusion: identification of the N-terminal region of A27L protein as the glycosaminoglycan-binding domain. *J Virol* **72**, 8374–9 (1998).
3. Lin, C. L., Chung, C. S., Heine, H. G. & Chang, W. Vaccinia virus envelope H3L protein binds to cell surface heparan sulfate and is important for intracellular mature virion morphogenesis and virus infection in vitro and in vivo. *J Virol* **74**, 3353–3365 (2000).
4. Gray, R. D. M. *et al.* Nanoscale polarization of the entry fusion complex of vaccinia virus drives efficient fusion. *Nat Microbiol* **4**, 1636–1644 (2019).
5. Wang, D. *et al.* Vaccinia Viral Protein A27 Is Anchored to the Viral Membrane via a Cooperative Interaction with Viral Membrane Protein. **289**, 6639–6655 (2014).
6. Howard, A. R., Senkevich, T. G. & Moss, B. Vaccinia Virus A26 and A27 Proteins Form a Stable Complex Tethered to Mature Virions by Association with the A17 Transmembrane Protein. *J Virol* **82**, 12384–12391 (2008).
7. Chang, T. H. *et al.* Crystal Structure of Vaccinia Viral A27 Protein Reveals a Novel Structure Critical for Its Function and Complex Formation with A26 Protein. *PLoS Pathog* **9**, 1–18 (2013).
8. Ching, Y.-C. *et al.* Disulfide Bond Formation at the C Termini of Vaccinia Virus A26 and A27 Proteins Does Not Require Viral Redox Enzymes and Suppresses Glycosaminoglycan-Mediated Cell Fusion. *J Virol* **83**, 6464–6476 (2009).
9. Vázquez, M. I., Esteban, M., Va, A. & Esteban, M. Identification of functional domains in the 14-kilodalton envelope protein (A27L) of vaccinia virus. *J Virol* **73**, 9098–9109 (1999).
10. Chang, H. *et al.* Vaccinia viral A26 protein is a fusion suppressor of mature virus and triggers membrane fusion through conformational change at low pH. *PLoS Pathog* **15**, 1–31 (2019).
11. Townsley, A. C. & Moss, B. Two Distinct Low-pH Steps Promote Entry of Vaccinia Virus. *J Virol* **81**, 8613–8620 (2007).
12. Rizopoulos, Z. *et al.* Vaccinia Virus Infection Requires Maturation of Macropinosomes. *Traffic* **16**, 814–831 (2015).
13. Chang, S., Chang, Y., Izmailyan, R., Tang, Y. & Chang, W. Vaccinia Virus A25 and A26 Proteins Are Fusion Suppressors for Mature Virions and Determine Strain-Specific Virus Entry Pathways into HeLa, CHO-K1, and L Cells. *J Virol* **84**, 8422–8432 (2010).
14. Mercer, J. & Helenius, A. Vaccinia Virus Uses Macropinocytosis and Apoptotic Mimicry to Enter Host Cells. *Science* (1979) **320**, 531–535 (2008).
15. Huang, C.-Y. *et al.* A novel cellular protein, VPEF, facilitates vaccinia virus penetration into HeLa cells through fluid phase endocytosis. *J Virol* **82**, 7988–99 (2008).

16. Mercer, J. *et al.* Vaccinia virus strains use distinct forms of macropinocytosis for host-cell entry. *Proceedings of the National Academy of Sciences* **107**, 9346–9351 (2010).
17. Niles, E. G. & Seto, J. *Vaccinia Virus Gene D8 Encodes a Virion Transmembrane Protein*. *JOURNAL OF VIROLOGY* vol. 62 (1988).
18. McMahon, H. T. & Gallop, J. L. Membrane curvature and mechanisms of dynamic cell membrane remodelling. *Nature* **428**, 590–596 (2005).

Dear Dr. Mercer,

Thank you for the submission of your revised manuscript to our editorial offices. I have now received the reports from the two referees that I asked to re-evaluate your study, you will find below. As you will see, both referees now fully support the publication of the study in EMBO reports.

Before I can proceed with formal acceptance, I have these editorial requests I ask you to address in a final revised manuscript:

- Please upload a completed author checklist with your final submission, which you can download from our author guidelines: <https://www.embopress.org/page/journal/14693178/authorguide>

Please insert page numbers in the checklist to indicate where the requested information can be found in the manuscript. The completed author checklist will also be part of the RPF.

- Please provide a final title with not more than 100 characters (including spaces).

- Please provide the abstract written in present tense throughout.

- Please provide sub-headings for the results section, rendering it more structured and comprehensive.

- Please add a paragraph titled 'Biosafety' to the methods section providing details on where and how biosafety-relevant experiments with viruses were performed and that these were approved, and by whom (institution, government).

- We updated our journal's competing interests policy in January 2022 and request authors to consider both actual and perceived competing interests. Please review the policy <https://www.embopress.org/competing-interests> and update your competing interests if necessary. Please name this section 'Disclosure and Competing Interests Statement' and put it after the Acknowledgements section.

- We now use CRediT to specify the contributions of each author in the journal submission system. CRediT replaces the author contribution section. Please use the free text box to provide more detailed descriptions and do not provide your final manuscript text file with an author contributions section. See also our guide to authors: <https://www.embopress.org/page/journal/14693178/authorguide#authorshipguidelines>

- Please order the manuscript sections like this, using these names:

Title page - Abstract - Keywords - Introduction - Results - Discussion - Materials and Methods - Data availability section - Acknowledgements - Disclosure and Competing Interests Statement - References - Figure legends - Expanded View Figure legends

- The Data Availability section should only contain information on large datasets that have been deposited to external repositories and all access information. If no primary datasets have been deposited, please state this here, e.g. using 'No primary datasets have been generated and deposited'. Please remove the sentence 'The datasets generated and/or analysed during the current study are available from the corresponding author on reasonable request'.

- Please make sure that the number "n" for how many independent experiments were performed, their nature (biological versus technical replicates), the bars and error bars (e.g. SEM, SD) and the test used to calculate p-values is indicated in the respective figure legends (for main, EV and Appendix figures) of the final revised manuscript. Please also check that all the p-values are explained in the legend, and that these fit to those shown in the figure. Please provide statistical testing where applicable. Please avoid the phrase 'independent experiment', but clearly state if these were biological or technical replicates. Please also indicate (e.g. with n.s.) if testing was performed, but the differences are not significant. In case n=2, please show the data as separate datapoints without error bars and statistics. See also:

<http://www.embopress.org/page/journal/14693178/authorguide#statisticalanalysis>

If n<5, please show single datapoints for diagrams. Presently, it seems that information related to n is missing in the legends of figures EV1d, f, g; EV2a, c.

- Please format the figure legends according to our journal style. See the respective section in our guide to authors (please find the link below). Please separate each panel description by a line brake and make sure that the panels are listed in alphabetic order. Moreover, please add to each legend a 'Data Information' section explaining the statistics used or providing information regarding replicates and scales.

- Could Figures EV2 and EV3 be fused?
- Please add scale bars also to Fig. EV4B.
- Please make sure that all the funding information is also entered into the online submission system and that it is complete and similar to the one in the acknowledgement section of the manuscript text file. Presently, a grant (?) by UCL Birkbeck is only mentioned in the acknowledgements.

In addition, I would need from you:

Best,

Referee #1:

It is my opinion that the authors have adequately addressed my comments from the initial round of review.

Referee #2:

The authors have responded appropriately to the reviewer's comments; their response is thoughtful and they have included additional information, discussion and references as needed.

The revised manuscript is improved, excellent and of significant interest for the field.

All editorial and formatting issues were resolved by the authors.

Prof. Jason Mercer
University College London
MRC-LMCB
Gower Street
Birmingham, LONDON B15 2TT
United Kingdom

Dear Prof. Mercer,

I am very pleased to accept your manuscript for publication in the next available issue of EMBO reports. Thank you for your contribution to our journal.

Yours sincerely,
